

# SISALv2: A comprehensive speleothem isotope database with multiple age-depth models

Laia Comas-Bru 1, Kira Rehfeld 2, Carla Roesch 2, Sahar Amirnezhad-Mozhdehi 3, Sandy P. Harrison 1, Kamolphat Atsawawaranunt 1,  Syed Masood Ahmad 4, Yassine Ait Brahim 5, Andy Baker 6, Matthew Bosomworth 1, Sebastian F.M. Breitenbach 7, Yuval Burstyn 8, Andrea Columbu 9, Michael Deininger 10, Attila Demény 11, Bronwyn Dixon 12, Jens Fohlmeister 13, István Gábor Hatvani 11, Jun Hu 14, Nikita Kaushal 15, Zoltán Kern 11, Inga Labuhn 16, Franziska A. Lechleitner 17, Andrew Lorrey 18, Belen Martrat 19, Valdir Felipe Novello 20, Jessica Oster 21, Carlos Pérez-Mejías 5, Denis Scholz 10, Nick Scroxton 22, Nitesh Sinha 23, Brittany Marie Ward 24, Sophie Warken 25, Haiwei Zhang 5 and SISAL Working Group members*.

Correspondence: Laia Comas-Bru (l.comasbru@reading.ac.uk)

**Affiliations:**

School of Archaeology, Geography, and Environmental Science, University of Reading, UK
Institute of Environmental Physics and Interdisciplinary Center for Scientific Computing, Heidelberg University, Germany
School of Geography. University College Dublin. Belfield, Dublin 4, Ireland
Department of Geography, Faculty of Natural Sciences, Jamia Millia Islamia, New Delhi, India
Institute of Global Environmental Change, Xi'an Jiaotong University, Xi'an, Shaanxi, China
Connected Waters Initiative Research Centre, UNSW Sydney, Sydney, New South Wales 2052, Australia
Department of Geography and Environmental Sciences, Northumbria University, Newcastle upon Tyne, UK
The Fredy & Nadine Herrmann Institute Earth Sciences, The Hebrew University of Jerusalem, The Edmond J. Safra Campus, Jerusalem 9190401, Israel
Department of Biological, Geological and Environmental Sciences (BiGeA), University of Bologna, Via Zamboni 67, 40126, Bologna, Italy
Institute for Geosciences, Johannes Gutenberg University Mainz, J.-J.-Becher-Weg 21, 55128 Mainz, Germany
Institute for Geological and Geochemical Research, Research Centre for Astronomy and Earth Sciences, H-1112, Budaörsi út 45, Budapest, Hungary
School of Geography, University of Melbourne, Australia; School of Archaeology, Geography, and Environmental Science, University of Reading, UK
Potsdam Institute for Climate Impact Research PIK, Potsdam, Germany
Department of Earth, Environmental and Planetary Sciences, Rice University, US
Asian School of the Environment, Nanyang Technological University, Singapore
Institute of Geography, University of Bremen, Celsiusstraße 2, 28359 Bremen, Germany
Department of Earth Sciences, South Parks Road, Oxford OX1 3AN, UK
National Institute of Water and Atmospheric Research, Auckland, 1010, New Zealand
Department of Environmental Chemistry, Spanish Council for Scientific Research (CSIC), Institute of Environmental Assessment and Water Research (IDAEA), Barcelona, Spain
Institute of Geoscience, University of São Paulo
Department of Earth and Environmental Sciences, Vanderbilt University, Nashville, TN 37240, US



School of Earth Sciences, University College Dublin, Belfield, Dublin 4, Ireland
IBS Center for Climate Physics (ICCP), Pusan National University, South Korea
Environmental Research Institute, University of Waikato, Hamilton, New Zealand
Institute of Earth Sciences and Institute of Environmental Physics, Heidelberg University, Germany
* A full list of authors appears at the end of the paper.

**Abstract:**
Characterising the temporal uncertainty in palaeoclimate records is crucial for analysing past climate
change, for correlating climate events between records, for assessing climate periodicities, identifying
potential triggers, and to evaluate climate model simulations. The first global compilation of speleothem
isotope records by the SISAL (Speleothem Isotope Synthesis and Analysis) Working Group showed that
age-model uncertainties are not systematically reported in the published literature and these are only
available for a limited number of records (ca. 15%, n = 107/691). To improve the usefulness of the SISAL
database, we have (i) improved the database's spatio-temporal coverage and (ii) created new
chronologies using seven different approaches for age-depth modelling. We have applied these
alternative chronologies to the records from the first version of the SISAL database (SISALv1) and to new
records compiled since the release of SISALv1. This paper documents the necessary changes in the
structure of the SISAL database to accommodate the inclusion of the new age-models and their
uncertainties as well as the expansion of the database to include new records and the quality-control
measures applied. This paper also documents the age-depth model approaches used to calculate the new
chronologies. The updated version of the SISAL database (SISALv2) contains isotopic data from 691
speleothem records from 294 cave sites and new age-depth models, including age-depth temporal
uncertainties for 512 speleothems. SISALv2 is available at http://dx.doi.org/10.17864/1947.242 (Comas-
Bru et al., 2020).
**1. Introduction**
Speleothems (secondary cave carbonates form from infiltrating rainwater after it percolates through the
soil, epikarst, and carbonate bedrock) are a rich terrestrial palaeoclimate archive. In particular, stable
oxygen and carbon isotopes ($\delta^{18}O$, $\delta^{13}C$) have been widely used to reconstruct regional and local
hydroclimate changes. The Speleothem Isotope Synthesis and Analyses (SISAL) Working Group is an
international effort, under the auspices of Past Global Changes (PAGES), to compile speleothem isotopic
records globally for the analysis of past climates (Comas-Bru and Harrison, 2019). The first version of the
SISAL database (Atsawawaranunt et al., 2018a; Atsawawaranunt et al., 2018b) contained 381 speleothem
records from 174 cave sites and has been used for analysing regional climate changes (Braun et al., 2019a;



Burstyn et al., 2019; Comas-Bru and Harrison, 2019; Deininger et al., 2019; Kaushal et al., 2018; Kern et
al., 2019; Lechleitner et al., 2018; Oster et al., 2019; Zhang et al., 2019). The potential for using the SISAL
database to evaluate climate models was explored using an updated version of the database (SISALv1b;
Atsawawaranunt et al., 2019) that contains 455 speleothem records from 211 sites (Comas-Bru et al.,

34   2019).

SISAL is continuing to expand the global database by including new records (Comas-Bru et al., 2020).
Although most of the records in SISALv2 (79.7%: Figure 1a) have been dated using the generally very
precise, absolute radiometric $^{230}$Th/U dating method, a variety of age-modelling approaches were
employed (Figure 1b) in constructing the original records. The vast majority of records provide no
information on the uncertainty of the age-depth relationship. However, many of the regional studies using
SISAL pointed the limited statistical power of analyses of speleothem records because of the lack of
temporal uncertainties. For example, these missing uncertainties prevented the extraction of underlying
climate modes during the last 2k years in Europe (Lechleitner et al., 2018). To overcome this limitation,
we have developed additional age-depth models for the SISALv2 records (Figure 2) in order to provide
robust chronologies with temporal uncertainties. The results of the various age-depth modelling
approaches differ because of differences in their underlying assumptions. We have used seven alternative
methods: linear interpolation, linear regression, Bchron (Haslett and Parnell, 2008), Bacon (Blaauw, 2010;
Blaauw and Christen, 2011; Blaauw et al., 2019), OxCal (Bronk Ramsey, 2008, 2009; Bronk Ramsey and
Lee, 2013), COPRA (Breitenbach et al., 2012) and StalAge (Scholz and Hoffmann, 2011). Comparison of
these different approaches provides a robust measure of the age uncertainty associated with any specific
speleothem record.
**2. Data and Methods**
**2.1 Construction of age-depth models: the SISAL chronology**
We attempted to construct age-depth models for 533 entities in an automated mode. For eight records,
this automated construction failed for all methods. For these records we provide manually constructed
chronologies, where no age model previously existed, and added a note in the database with details on
the construction procedure. Age models for 21 records were successfully computed but later dropped in
the screening process due to inconsistent information or incompatibility for an automated routine. In
total, we provide a new chronology for 512 speleothem records in SISALv2.
The SISAL chronology provides alternative age-depth models for SISAL records that are not composites
(i.e., time-series based on more than one speleothem record), that have not been superseded in the



database by a newer entity and which are purely $^{230}$Th/U dated. We therefore excluded records for which
the chronology is based on lamina counting, radiocarbon ages or a combination of methods. This decision
was based on the low uncertainties of the age-depth models based on lamina counting and the challenge
of reproducing age-depth models based on radiocarbon ages. We made an exception with the case of
entity_id 163 (Talma et al., 1992), which covers two key periods, the Mid-Holocene and the Last Glacial
Maximum, at high temporal resolution. In this case, we calculated a new SISAL chronology based on the
provided $^{230}$Th/U dates but did not consider the uncorrected $^{14}$C ages upon which the original age-depth
model is based. We also excluded records for which isotopic data is not available (i.e., entities that are
part of composites) and entities that are constrained by less than three dates. Additionally, the dating
information for 23 entities shows hiatuses at the top/bottom of the speleothem that are not constrained
by any date. For these records, we partially masked the new chronologies to remove the unconstrained
section(s). Original dates were used without modification in the age-depth modelling.
To allow a comprehensive cross-examination of uncertainties, seven age-depth modelling techniques
were implemented here across all selected records. Due to the high number of records (n = 533), all
methods were run in batch mode. A preliminary study, using the database version v1b demonstrated the
feasibility of the automated construction and evaluation of age-depth models using a subset of records
and methods (Roesch and Rehfeld, 2019). Further details on the evaluation of the updated age-depth
models are provided in Section 3.2. The seven different methods are briefly described below. All methods
assume that growth occurred along a single growth axis. For one entity, where it was previously known
that two growth axes exist, we added an explanatory statement in the database. All approaches except
StalAge produce Monte Carlo (MC) iterations of the age-depth models. We provide 1,000 MC iterations
for each new SISALv2 chronology (https://doi.org/10.5281/zenodo.3591197).
Major challenges arise through hiatuses (growth interruptions) and age reversals. In the classification of
the reversals, we distinguish between tractable reversals (with overlapping confidence intervals) and non-
tractable reversals (i.e., where the two-sigma-dating uncertainties do not overlap) following the definition
of Breitenbach et al. (2012). We developed a workflow to treat records with hiatuses (Roesch and Rehfeld,
2019; details below), which allowed the construction of age-depth models for 20% of the records with
one or more hiatuses. Changes, such as the hiatus treatment and outlier age modification, are recorded
in a logfile created when running the age models. We followed the original author's choices with regard
to date usage. If an age was marked as "not used" or "usage unknown", we did not consider this in the
construction of the new chronologies except in OxCal, where dates with "usage unknown" were
considered.





1) **Linear Interpolation** (*lin_interp_age*) between radiometric dates. This is the classic approach for age-
depth model construction for palaeoclimate archives and was used in 32.1% of the original age-depth
models in SISALv2. Here, we extend this approach and calculate the age uncertainty by sampling the range
of uncertainty of each $^{230}$Th/U-age, 2,000 times, assuming a Gaussian distribution. This is consistent with
the implementation of linear interpolation in CLAM (Blaauw, 2010) and COPRA (Breitenbach et al., 2012).
Linear interpolation was implemented in R (R Core Team, 2019), using the `approxExtrap()` function
in the `Hmisc` package. We included an automated reversal check that increases the dating uncertainties
until a monotonic age model is achieved, similar to that of StalAge (Scholz and Hoffmann, 2011). Hiatuses
are modelled following the approach of Roesch and Rehfeld (2019), where rather than modelling each
segment separately, synthetic ages with uncertainties spanning the entire hiatus duration are introduced
for use in age-depth model construction. These synthetic ages are removed after age-depth model
construction. Linear interpolation was applied to 80% (n=408/512) of the SISAL records for which new
chronologies were developed.
2) **Linear Regression** *(lin_reg_age)* provides a single best fit line through all available radiometric ages
assuming a constant growth rate. Linear regression was used in 6.7% of the original SISALv2 age models.
As with linear interpolation, age uncertainties are based on randomly sampling the U-series dates to
produce 2,000 age-depth models (i.e., ensembles). Temporal uncertainties are then given by the
uncertainty of the median-based fit to each ensemble member. If hiatuses are present, the segments in-
between were split at the depth of the hiatus without an artificial age. The method is implemented in R,
using the `lm()` function from the base package. Linear regression was applied to 36% (n=185/512) of the
SISAL records for which new chronologies were developed.
3) **Bchron** *(Bchron_age)* is a Bayesian method based on a continuous Markov processes (Haslett and
Parnell, 2008) and available as an R package (Parnell, 2018). This method was originally used for only one
speleothem record in SISALv2. Since *Bchron* cannot handle hiatuses, we implemented a new workflow
that adds synthetic ages with uncertainties spanning the entire hiatus duration (Roesch and Rehfeld,
2019), as performed with linear interpolation, StalAge and our implementation of COPRA. Bchron provides
age-depth model ensembles of which we have kept the last 2,000. Here we use the function `bchron()`
with `jitter.positions = true` to mitigate problems due to rounded-off depth values. This
method has been applied to 83% (n=426/512) of the SISAL records for which new chronologies were
developed.
4) **Bacon** (*Bacon_age*) is a semi-parametric Bayesian method based on autoregressive gamma-processes
(Blaauw, 2010; Blaauw and Christen, 2011; Blaauw et al., 2019). It was used in three of the original
chronologies in SISALv2. The R package *rBacon* can handle both outliers and hiatuses and apart from



giving the median age-depth model, it also returns the Monte Carlo realisations (i.e. ensembles), from
which the median age-depth model is calculated. During the creation of the SISAL chronologies, the
existing *rBacon* package (version 2.3.9.1) was updated to improve the handling of stalagmite growth rates
and hiatuses. We use this revised version, available on CRAN ([https://cran.r-](https://cran.r-)
[project.org/web/packages/rbacon/index.html](https://cran.r-project.org/web/packages/rbacon/index.html)), to provide a median age-depth model and an ensemble
of age-model realisations for 65% (n=335/512) of the SISAL records for which new chronologies were
developed.
5) **OxCal** *(Oxcal_age)* is a Bayesian chronological modelling tool that uses Markov Chain Monte Carlo
(Bronk Ramsey, 2009). This method was used in 4.1% of the original SISALv2 chronologies. OxCal can deal
with hiatuses and outliers and accounts for the non-uniform nature of the deposition process (Poisson
process using the P_Sequence command). Here we used the analysis module of OxCal version 4.3 with a
default initial value of interpolation rate of 1 and an initial value of model rigidity (k) of $k_0=1$ with a uniform
distribution from 0.01 to 100 for the range of $k/k_0$ ($log10(k/k_0)=(-2,2)$) (C. Bronk Ramsey, personal
communication). The initial value of the interpolation rate determines the number of points between any
two dates, for which an age will be calculated. We subsequently linearly interpolated the age-depth model
to the depths of individual isotope measurements. Were multiple dates are given for the same depth for
any given entity, the date with the smallest uncertainty was used to construct the SISAL chronology. In
case of asymmetric uncertainties in the dating table, the largest uncertainty value was chosen. We kept
the last 2,000 realisations of the age-depth models for each entity. OxCal chronologies are available for
21% (n=106/512) of the SISAL records for which new chronologies were developed.
6) **COPRA** (co*pRa_age)* is an approach based on interpolation-between-dates (Breitenbach et al., 2012)
and was used for 9.7% of the original SISALv2 chronologies. COPRA is available as a Matlab package with
a graphical user interface (GUI) that has interactive checks for reversals and hiatuses. The Matlab version
can handle multiple hiatuses and (to some extent) layer-counted segments. However, age-reversals can
occur near short-lived hiatuses. To overcome this, we implemented a new workflow in R that adds
artificial dates at the location of the hiatuses and prevents the creation of age reversals (Roesch and
Rehfeld, 2019) as done with linear interpolation, StalAge and Bchron. Additionally, we also incorporated
an automated reversal check similar to that already embedded into StalAge (Scholz and Hoffmann, 2011).
This R version, copRa, uses the default piecewise-cubic-hermite-interpolation (pchip) algorithm in R
without consideration of layer counting. This approach was used for 76% (n= 389/512) of the SISAL records
for which new chronologies were developed.
7) **StalAge** *(StalAge_age)* fits straight lines through three adjacent dates using weights based on the dating
measurement errors (Scholz and Hoffmann, 2011). Age uncertainties are iteratively obtained through a





Monte Carlo approach, but ensembles are not given in the output. StalAge was used to construct 13.1%
of the original SISALv2 chronologies. The StalAge v1.0 R function has been updated to R version 3.4 and
the default outlier and reversal checks were enabled to run automatically. Hiatuses cannot be entered in
StalAge v1.0, but the updated version incorporates a treatment of hiatuses based on the creation of
temporary synthetic ages following Roesch and Rehfeld (2019). In contrast to other methods, mean ages
instead of median ages are reported for StalAge. StalAge was applied to 62% (n=320/512) of the SISAL
records for which new chronologies were developed.

**2.2 Revised structure of the database**

The data are stored in a relational database (MySQL), which consists of 15 linked tables: *site, entity,*
*sample, dating, dating_lamina, gap, hiatus, original_chronology, d13C, d18O, entity_link_reference,*
*references, composite_link_entity, notes* and *sisal_chronology*. Figure 3 shows the relationships between
these tables and the type of each field (e.g. numeric, text). The structure and contents of all tables except
the new *sisal_chronology* table are described in detail in Atsawawaranunt et al. (2018a). Here, we focus
on the new *sisal_chronology* table and on the changes that were made to other tables in order to
accommodate this new table (See section 2.3). Details of the fields in this new table are listed in Table 1.
Changes were also made to the dating table (*dating*) to accommodate information about whether a
specific date was used to construct each of the age-depth models in the *sisal_chronology* table (Table 2).
We followed the original authors' decision regarding the exclusion of dates (i.e. because of high
uncertainties, age reversals or high detrital content). However, some dates used in the original age-depth
model were not used in the SISALv2 chronologies to prevent unrealistic age-depth relationships (i.e. age
inversions). Information on whether a particular date was used for the construction of specific type of
age-depth model is provided in the dating table, under columns labelled *date_used_lin_interp,*
*date_used_lin_reg, date_used_Bchron, date_used_Bacon, date_used_OxCal, date_used_copRa* and
*date_used_StalAge* (Table 2).
The dating and the sample tables were modified to accommodate the inclusion of new entities in the
database. Specifically, the pre-defined options lists were expanded, options that had never been used
were removed, and some typographical errors in the field names were corrected; these changes are listed
in Table 3.

## 3. Quality Control

### 3.1 Quality control of individual speleothem records

The quality control procedure for individual records newly incorporated in the SISALv2 database is based on the steps described in Atsawawaranunt et al. (2018a). We have updated the Python database scripts to provide a more thorough quality assessment of individual records. Additional checks of the dating table resulted in modifications in the *230Th_232Th, 230Th_238U, 234U_238U, ini230Th_232Th, 238U_content, 230Th_content, 232Th_content* and *decay constant* fields in the dating table for 60 entities. A summary of the fields that are both automatically and manually checked before uploading a record to the database is available in Appendix 1.

Analyses of the data included in SISALv1 (Braun et al., 2019a; Burstyn et al., 2019; Deininger et al., 2019; Kaushal et al., 2018; Kern et al., 2019; Lechleitner et al., 2018; Oster et al., 2019; Zhang et al., 2019) and SISALv1b (Comas-Bru et al., 2019) revealed a number of errors in specific records that have now been corrected. These revisions include, for example, updates in mineralogies (*sample.mineralogy*), revised coordinates (*site.latitude* and/or *site.longitude*) and addition of missing information that was previously entered as "unknown". The fields affected and the number of records with modifications are listed in Table 4. All revisions are also documented at Comas-Bru et al., 2020.

### 3.2 Quality control of the age-depth models in the SISAL chronology

The conception and the test of the R workflow, integrating all methods but OxCal, was outlined in Roesch and Rehfeld (2019) and includes automatized checks for the final chronologies except for OxCal. The quality control parameters obtained from OxCal were compared with the recommended values of Agreement Index (A) > 60% and Convergence (C) > 95%, in accordance with the guidelines in Bronk Ramsey (2008). In addition to both model agreement and P_Sequence convergence meeting these criteria, at least 90% of individual dates had to have an acceptable Agreement and Convergence themselves. OxCal age-depth models failing to meet these criteria were not included in the SISAL chronology table.

An overview of the evaluation results for the age-depth models constructed in automated mode is given in Figure 4. Three nested criteria are used to evaluate them. Firstly, chronologies with reversals (Check 1) are automatically rejected (score -1). Secondly, the final chronology should flexibly follow clear growth rate changes (Check 2), such that 70% of the dates are encompassed in the final age-depth model within 4 sigma uncertainty (score +1). Thirdly, temporal uncertainties are expected to increase between dates and near hiatuses (Check 3). This criterion is met in the automated screening (score +1) if the Interquartile range (IQR) is higher between dates or at hiatuses than at dates. Only entities that pass all three criteria





are considered successful. All age-depth models that satisfied Check 1 were also evaluated in an expert-
based manual screening by ten people. If more than two experts agreed that an individual age-depth
model was unreliable or inconsistencies, such as large offsets between the original age model and the
dates marked as 'used', occurred, the model was not included in the SISAL chronology table. This
automatic and expert-based quality control screening resulted in 2,138 new age-depth models
constructed for 503 SISAL entities.
**4. Recommendation for the use of SISAL chronologies**
The original age-depth models for every entity are available in SISALv2. However, given the lack of age
uncertainties for most of the records, we recommend considering the SISAL chronologies with their
respective 95% confidence intervals whenever possible. No single age-depth modelling approach is
successful for all entities, and we therefore recommend that all the methods for a specific entity are used
together in visual and/or statistical comparisons. Depending on methodological choices, age-depth
models compatible with the dating evidence can result in considerable temporal differences for
transitions (Figure 5). For analyses relying on the temporal alignment of records (e.g. cross-correlation),
age-depth model uncertainties should be considered using the ensemble of compatible age-depth models
as described, e.g., in Mudelsee et al. (2012), Rehfeld and Kurths (2014) and Hu et al. (2017).
**5. Overview of database contents**
SISALv2 contains 353.976 $\delta^{18}$O and 200,613 $\delta^{13}$C measurements from 673 individual speleothem records
and 18 composite records from 293 cave systems (Table 5; Figure 2; Comas-Bru et al., 2020). There are 20
records included in SISALv2 that are identified as being superseded and linked to the newer records; their
original datasets are included in the database for completeness. This is an improvement of 235 records
from SISALv1b (Atsawawaranunt et al., 2019; Comas-Bru et al., 2019; Table 6). SISALv2 represents 72% of
the existing speleothem records identified by the SISAL Working Group and more than three times the
number of speleothem records in the NCEI-NOAA repository (n = 210 as of November 2019), which is the
one most commonly used by the speleothem community to make their data publicly available. SISALv2
also contains nine records that have not been published or are only available in PhD theses.
The published age-depth models of all speleothems are accessible in the *original_chronology* metadata
table and our standardised age-depth models are available at the *sisal_chronology* table for 512
speleothems. Temporal uncertainties are now provided for 79% of the records in the SISAL database.
This second version of the SISAL database has an improved spatial coverage compared to SISALv1
(Atsawawaranunt et al., 2018b) and SISALv1b (Figure 3; Atsawawaranunt et al., 2019). SISALv2 contains



most published records from Oceania (80.2%), Africa (73.7%) and South America (77.6%), but
improvements are still possible in regions like the Middle East (42.3%) and Asia (64.8%) (Table 6).
The temporal distribution of records for the past 2,000 years is good, with 181 speleothems covering at
least one-third of this period and 84 records covering the entire last 2k (-68 to 2,000 years BP) with an
average resolution of 20 isotope measurements in every 100-year slice (Figure 6a). There are 182 records
that cover at least one-third of the Holocene (last 11,700 years BP) with 37 of these covering the whole
period with at least one isotope measurement in every 500-year period (Figure 6b). There are 84 entities
during the deglaciation period (21,000 to 11,700 years BP) with at least one measurement in every 500-
year time period (Figure 6b). The Last Interglacial (130,000 to 115,000 years BP) is covered by 47
speleothem records that record at least one-third of this period with, on average, 25 isotope
measurements at every 1,000-year time-slice (Fig. 6c).
This updated SISALv2 database now provides the basis not only for comparing a large number of
speleothem-based environmental reconstructions on regional to a global scale, but also allows for
comprehensive analyses of stable isotope records on various timescales from multi-decadal to orbital.
**6. Data and code availability:**
The database is available in SQL and CSV format from http://dx.doi.org/10.17864/1947.242 (Comas-Bru
et al., 2020). The code used for constructing the linear interpolation, linear regression, Bchron, Bacon,
copRa and StalAge age-depth models is available at https://github.com/palaeovar/SISAL.AM. *rBacon*
package (version 2.3.9.1) is available on CRAN (https://cran.r-
project.org/web/packages/rbacon/index.html). The code used to construct the OxCal age-depth models
and trim the ensembles output to the last 2,000 iterations is available at
https://doi.org/10.5281/zenodo.3586280. The ensembles are available at
https://doi.org/10.5281/zenodo.3591197. The workbook used to submit data to SISAL is available as a
supplementary document of Comas-Bru and Harrison (2019); also available at
https://10.5281/zenodo.3631403. The codes for the quality control assessment of the data submitted to
SISAL can be obtained from https://10.5281/zenodo.3631403. The codes to assess the dating table in
SISALv2 are available at https://github.com/jensfohlmeister/QC_SISALv2_dating_metadata and
https://10.5281/zenodo.3631443. Details on the Quality Control assessments are available in the
Supplementary material.



**Author contributions:**

LCB is the coordinator of the SISAL working group. LCB, SPH and KR designed the new version of the database. KR coordinated the construction of the new age-depth models except OxCal. All age-depth models except OxCal were run by CR and KR. LCB coordinated the construction of the OxCal age-depth models, which were run by SAM and LCB. LCB implemented the changes in the v2 of the database with the assistance of KA. SMA, YAB, AB, YB, MB, AC, MD, AD, BD, IGH, JH, NK, ZK, FAL, AL, BM, VFN, JO, CPM, NS, NS, BMW, SW and HZ coordinated the regional data collection and the age-model screening. SFMB, MB and DS provided support for COPRA, Bacon and StalAge, respectively. JF assisted in the Quality Control procedure of the SISAL database. Figures 1, 4 and 5 were created by CR and KR. Figures 2, 3 and 6 were created by LCB. All authors listed as "SISAL Working Group members" provided data for this version of the database and/or helped to complete data entry. The first draft of the paper was written by LCB with inputs by KR and SPH and all authors contributed to the final version.

**Team list:**

The following SISAL Working Group members contributed with data to SISALv2: James Apaéstegui (Instituto Geofísico del Perú, Lima, Peru), Lisa M. Baldini (School of Health & Life Sciences, Teesside University, Middlesbrough, UK), Shraddha Band (Geoscience Department, National Taiwan University, No.1, Sec. 4, Roosevelt Road, Taipei 106, Taiwan), Maarten Blaauw (School of Natural and Built Environment, Queen's University Belfast, U.K.), Ronny Boch (Institute of Applied Geosciences, Graz University of Technology, Rechbauerstraße 12,8010 Graz, Austria, Andrea Borsato (School of Environmental and Life Sciences, University of Newcastle, Challaghan 2308, NSW, Australia), Alexander Budsky (Institute for Geosciences, Johannes Gutenberg University Mainz, Johann-Joachim-Becher-Weg 21, 55128 Mainz, Germany), Maria Gracia Bustamante Rosell (Department of Geology and Environmental Science, University of Pittsburgh, USA), Sakonvan Chawchai (Department of Geology, Faculty of Science, Chulalongkorn University, Bangkok 10330, Thailand), Silviu Constantin (Emil Racovita Institute of Speleology, Bucharest, Romania and Centro Nacional de Investigación sobre la Evolución Humana, CENIEH, Burgos, Spain), Rhawn Denniston (Department of Geology, Cornell College, Mount Vernon, IA 52314, USA), Virgil Dragusin (Emil Racovita Institute of Speleology, 010986, Strada Frumoasă 31, Bucharest, Romania), Russell Drysdale (School of Geography, University of Melbourne, Melbourne, Australia), Oana Dumitru (Karst Research Group, School of Geosciences, University of South Florida, 4202 E. Fowler Ave., NES 107, Tampa, FL 33620, USA), Amy Frappier (Department of Geosciences, Skidmore College, Saratoga Springs, New York, USA), Naveen Gandhi (Indian Institute of Tropical Meteorology, Dr. Homi Bhabha Road, Pashan, Pune-411008, India), Pawan Gautam (Centre for Earth, Ocean and



Atmospheric Sciences, University of Hyderabad, India; now at Geological Survey of India, Northern Region,
India), Li Hanying (Institute of Global Environmental Change, Xi'an Jiaotong University, China), Ilaria Isola
(Istituto Nazionale di Geofisica e Vulcanologia, Pisa, Italy), Xiuyang Jiang (College of Geography Science,
Fujian Normal University, Fuzhou 350007, China), Zhao Jingyao (Institute of Global Environmental Change,
Xi'an Jiaotong University, China), Kathleen Johnson (Dept. of Earth System Science, University of
California, Irvine, 3200 Croul Hall, Irvine, CA 92697 USA), Vanessa Johnston (Research Centre of Slovenian
Academy of Sciences and Arts ZRC SAZU, Novi trg 2, Ljubljana, Slovenia), Gayatri Kathayat (Institute of
Global Environmental Change, Xi'an Jiaotong University, China), Jennifer Klose (Institut für
Geowissenschaften, Johannes Gutenberg-Universität Mainz, Germany),   Claire Krause (Geoscience
Australia, Canberra, Australian Capital Territory, 2601, Australia), Matthew Lachniet (Department of
Geoscience, University of Nevada Las Vegas, Las Vegas, NV 89154), Amzad Laskar (Geosciences Division,
Physical Research Laboratory, Navrangpura, Ahmedabad 380009, India), Stein-Erik Lauritzen (University
of Bergen, Earth science, Norway), Nina Loncar (University of Zadar, Department of Geography, 23000,
Ulica Mihovila Pavlinovića, Zadar, Croatia), Gina Moseley (Institute of Geology, University of Innsbruck,
Innrain 52, 6020 Innsbruck, Austria), Allu C Narayana (Centre for Earth, Ocean and Atmospheric Sciences,
University of Hyderabad, India), Bogdan P. Onac (University of South Florida, School of Geosciences, 4202
E Fowler Ave, Tampa, FL 33620, USA, Emil Racovita Institute of Speleology, Cluj-Napoca, Romania), Jacek
Pawlak (Institute of Geological Sciences, Polish Academy of Sciences, 00-818, Twarda 51/55 , Warsaw,
Poland), Christopher Bronk Ramsey (Research Laboratory for Archaeology and the History of Art, Oxford
University, Oxford, UK), Isabel Rivera-Collazo (Department of Anthropology and the Scripps Institution of
Oceanography, UC San Diego, USA), Carlos Rossi (Dept. Petrología y Geoquímica, Facultad de Ciencias
Geologicas, Universidad Complutense, Madrid, Spain), Peter J. Rowe (School of Environmental Sciences,
University of East Anglia, NR4 7TJ, Norwich Research Park, Norwich, UK), Nicolás M. Stríkis (Department
of Geochemistry, Universidade Federal Fluminense, Niterói, Brazil), Liangcheng Tan (State Key Laboratory
of Loess and Quaternary Geology, Institute of Earth Environment, Chinese Academy of Sciences, Xi'an
710075, China), Sophie Verheyden (Politique scientifique fédérale belge BELSPO, Bvd. Simon Bolivar
30,1000 Brussels), Hubert Vonhof (Max Planck Institute for Chemistry, Mainz, Germany), Michael Weber
(Johannes    Gutenberg-Universität    Mainz,    Germany),    Kathleen    Wendt    (Geo-    und
Atmosphärenwissenschaften, Universität Innsbruck, Austria), Paul Wilcox (Institute of Geology, University
of Innsbruck, Austria), Amos Winter (Dept. of Earth and Environmental Systems, Indiana State University,
USA), Jiangying Wu (School of Geography, Nanjing Normal University, Nanjing, China), Peter Wynn
(Lancaster Environment Centre, University of Lancaster, Lancaster, LA1 4YQ UK), Madhusudan G. Yadava
(Geosciences Division, Physical Research Laboratory, Navrangpura, Ahmedabad 380009, India).

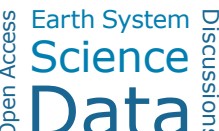

**Competing Interests:**
The authors declare no competing interests.
**Funding:**
SISAL (Speleothem Isotopes Synthesis and Analysis) is a working group of the Past Global Changes (PAGES)
programme. We thank PAGES for their support for this activity. The design and creation of v2 of the
database was supported by funding to SPH from the ERC-funded project GC2.0 (Global Change 2.0:
Unlocking the past for a clearer future, grant number 694481) and the Geological Survey Ireland Short Call
2017 (Developing a toolkit for model evaluation using speleothem isotope data, grant number 2017-SC-
056) award to LCB. SPH and LCB acknowledge additional support from the ERC-funded project GC2.0 and
from the JPI-Belmont project "PAlaeo-Constraints on Monsoon Evolution and Dynamics (PACMEDY)"
through the UK Natural Environmental Research Council (NERC). KR and DS acknowledge support by the
Deutsche Forschungsgemeinschaft (DFG, codes RE3994/2-1 and SCHO 1274/11-1).
**Acknowledgements**
SISAL (Speleothem Isotopes Synthesis and Analysis) is a working group of the Past Global Changes (PAGES)
programme. We thank PAGES for their support for this activity. We thank SISAL members who
contributed their published data to the database and provided additional information when necessary.
We thank all experts who engaged in the age-depth model evaluation. The authors would like to
acknowledge Avner Ayalon, Jordi López, Bahadur Singh Kotlia, Dennis Rupprecht.

**List of Figures and Tables**

**Figure 1:** Summary of the dating information on which the original age-depth models are based (a) and the original age-depth model types (b) present in SISALv2.

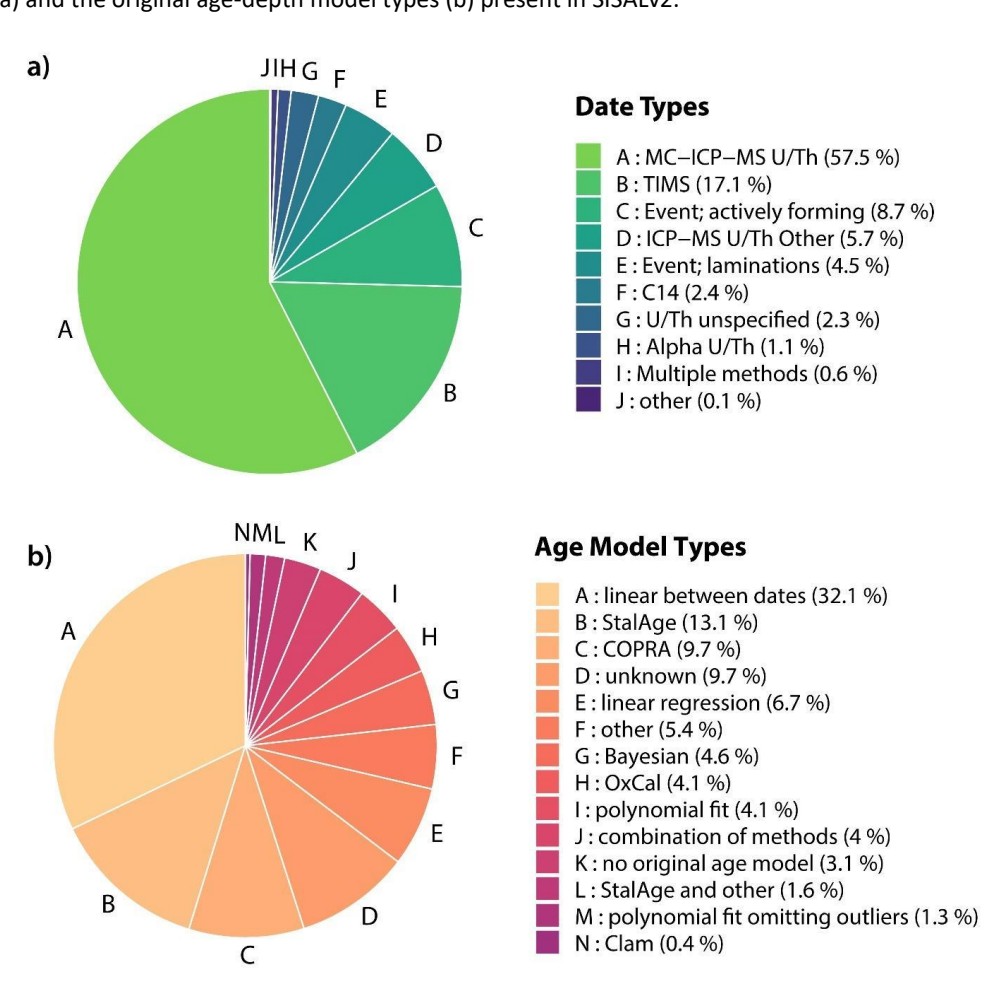



**Figure 2:** Cave sites included in the version 1, 1b and 2 of the SISAL database on the Global Karst
Aquifer Map (WOKAM project; Chen et al., 2017: https://www.un-igrac.org/resource/world-
karst-aquifer-map-wokam).

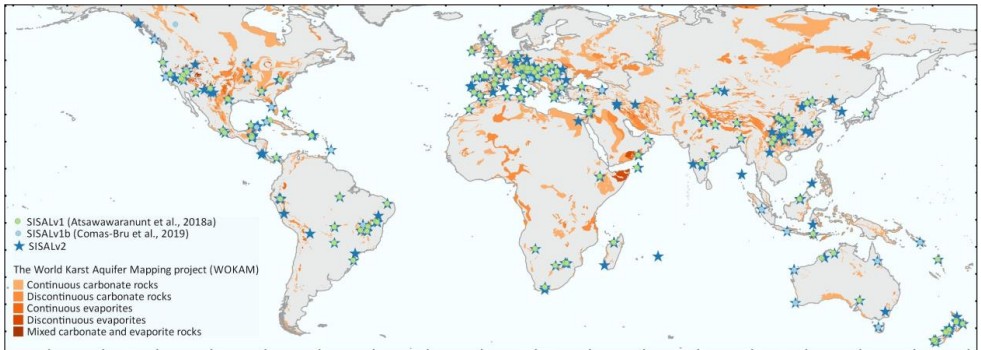



**Figure 3:** The structure of the SISAL database version 2. Fields and table marked with (*) refer to
new information added to SISALv1b; see tables 1 and 2 for details. The colours refer to the format
of that field: Enum, Int, Varchar, Double or Decimal. More information on the list of pre-defined
menus can be found in Atsawawaranunt et al. (2018a).

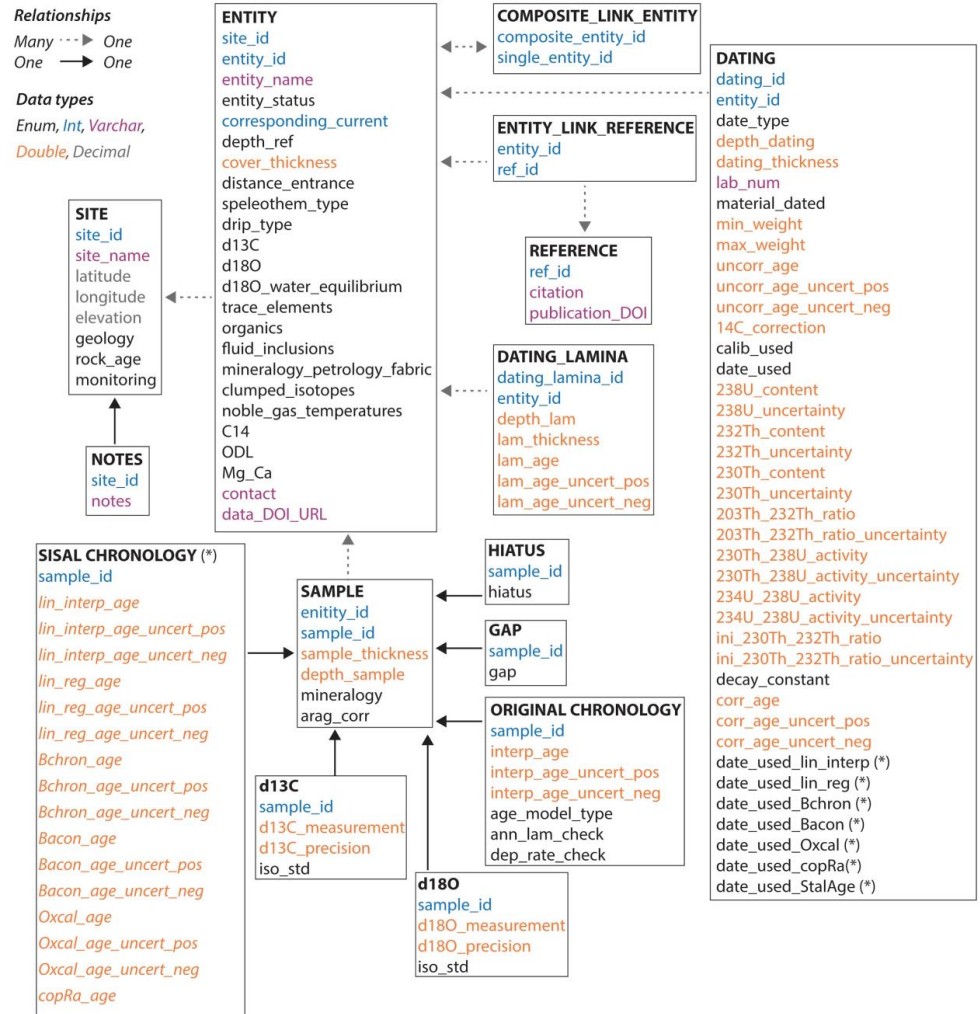




**Figure 4:** Visual summary of quality control of the automated SISAL chronology construction. The evaluation of the age-depth models for each method (x-axis) is given for each entity (y-axis) that was considered for the construction (n=533). Black lines mark age-depth models that could not be computed. Age-depth models dropped in the automated or expert evaluation are marked by grey lines. Age-depth models retained in SISALv2 are scored from 1 (only one criterion satisfied) to 3 (all criteria satisfied) in shades of blue. For 504 records alternative age-depth models with uncertainties are provided (green lines) in the "success" column.

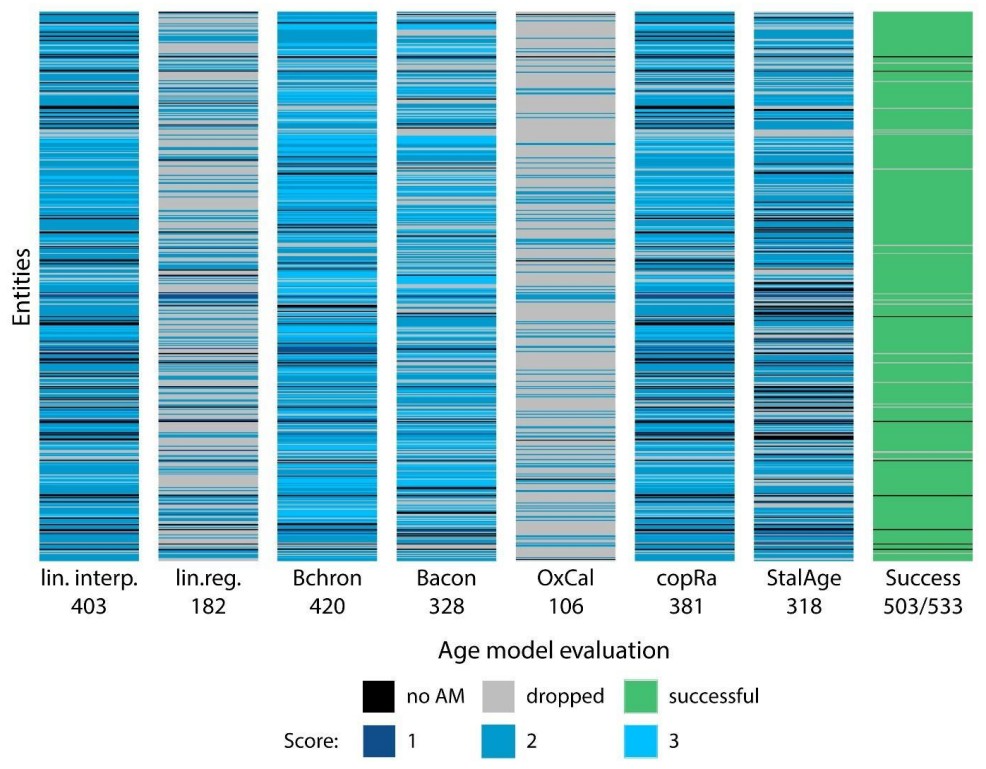

**Figure 5:** Illustration of the impact of the age model choice on reconstructed speleothem
chronology illustrated by the KNI-51-H speleothem record (entity_id 342; Denniston et al.,
2013b). Panel (a) shows the median and mean age estimates for each downcore sample from
the different age models; (b) shows the interquartile range (IQR) of the ages. Horizontal dashed
lines show the depths of the measured dates; (c) shows the isotopic record using the different
age models.

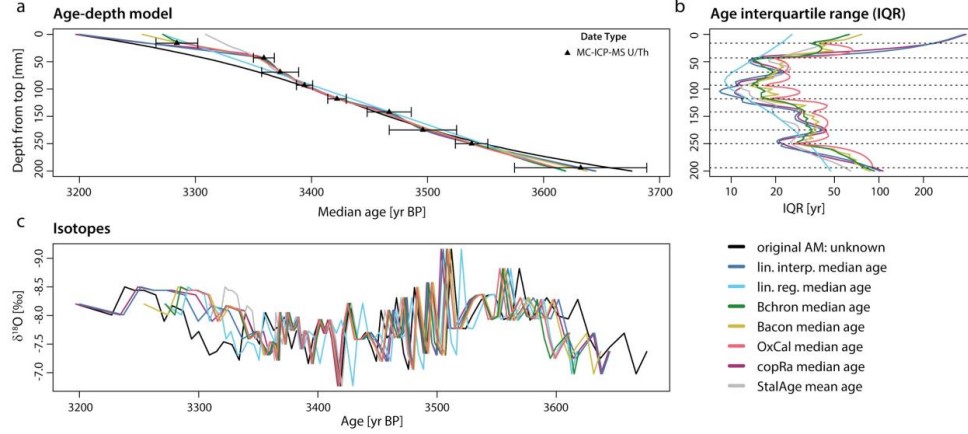


**Figure 6:** Global and regional temporal coverage of entities in the SISALv2. (a) last 2,000 years
with a bin size of 10 years; (b) last 21,000 years with a bin size of 500 years; (c) the period between
115,000 and 130,000 years BP with a bin size of 1,000 yrs. BP refers to "Before Present" where
present is 1950 CE. Regions defined as in Table 7.

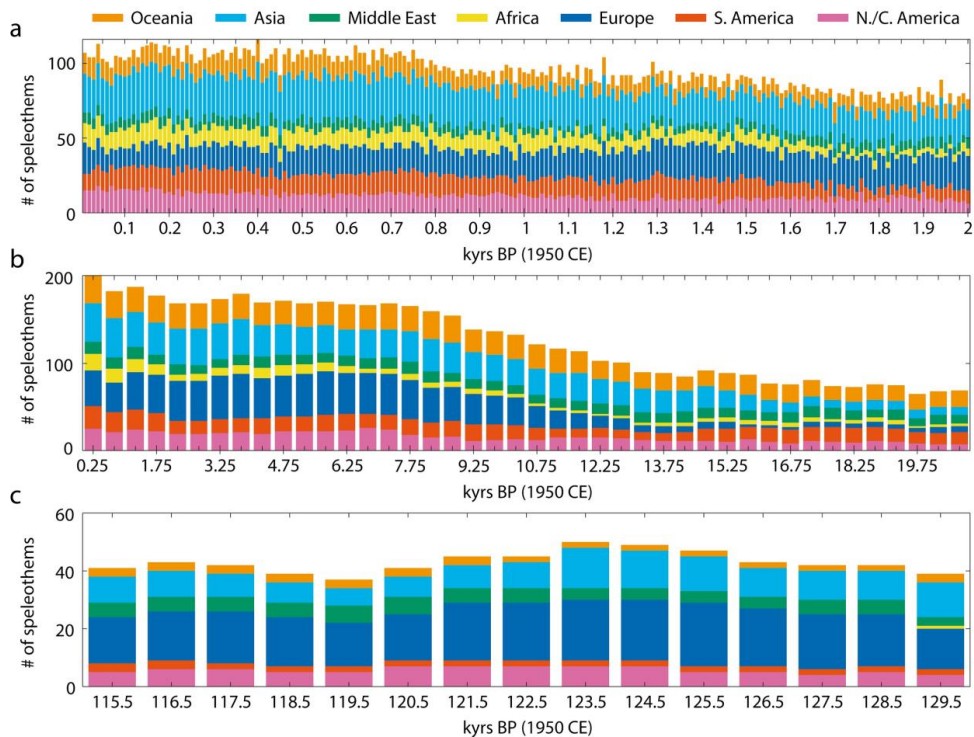



**Table 1:** Details of the sisal_chronology table. All ages in SISAL are reported as years BP (Before Present) where present is 1950 CE.

| Field label | Description | Format | Constraints |
|---|---|---|---|
| *sample_id* | Refers to the unique identifier for the sample (as given in the sample table) | Numeric | Positive integer |
| *lin_interp_age* | Age of the sample in years calculated with linear interpolation between dates | Numeric | None |
| *lin_interp_age_uncert_pos* | Positive 2-sigma uncertainty of the age of the sample in years calculated with linear interpolation between dates | Numeric | Positive decimal |
| *lin_interp_age_uncert_neg* | Negative 2-sigma uncertainty of the age of the sample in years calculated with linear interpolation between dates | Numeric | Positive decimal |
| *lin_reg_age* | Age of the sample in years calculated with linear regression | Numeric | None |
| *lin_reg_age_uncert_pos* | Positive 2-sigma uncertainty of the age of the sample in years calculated with linear regression | Numeric | Positive decimal |
| *lin_reg_age_uncert_neg* | Negative 2-sigma uncertainty of the age of the sample in years calculated with linear regression | Numeric | Positive decimal |
| *Bchron_age* | Age of the sample in years calculated with Bchron | Numeric | None |
| *Bchron _age_uncert_pos* | Positive 2-sigma uncertainty of the age of the sample in years calculated with Bchron | Numeric | Positive decimal |
| *Bchron _age_uncert_neg* | Negative 2-sigma uncertainty of the age of the sample in years calculated with Bchron | Numeric | Positive decimal |
| *Bacon_age* | Age of the sample in years calculated with Bacon | Numeric | None |
| *Bacon _age_uncert_pos* | Positive 2-sigma uncertainty of the age of the sample in years calculated with Bacon | Numeric | Positive decimal |
| *Bacon_age_uncert_neg* | Negative 2-sigma uncertainty of the age of the sample in years calculated with Bacon | Numeric | Positive decimal |
| *OxCal_age* | Age of the sample in years calculated with OxCal | Numeric | None |
| *OxCal_age_uncert_pos* | Positive 2-sigma uncertainty of the age of the sample in years calculated with OxCal | Numeric | Positive decimal |
| *OxCal_age_uncert_neg* | Negative 2-sigma uncertainty of the age of the sample in years calculated with OxCal | Numeric | Positive decimal |
| *copRa_age* | Age of the sample in years calculated with copRa | Numeric | None |
| *copRa _age_uncert_pos* | Positive 2-sigma uncertainty of the age of the sample in years calculated with copRa | Numeric | Positive decimal |



| copRa_age_uncert_neg | Negative 2-sigma uncertainty of the age of the sample in years calculated with copRa | Numeric | Positive decimal |
|---|---|---|---|
| Stalage_age | Age of the sample in years calculated with StalAge | Numeric | None |
| Stalage_age_uncert_pos | Positive 2-sigma uncertainty of the age of the sample in years calculated with StalAge | Numeric | Positive decimal |
| Stalage_age_uncert_neg | Negative 2-sigma uncertainty of the age of the sample in years calculated with StalAge | Numeric | Positive decimal |


**Table 2:** Changes made to the Dating table to accommodate the new age models. These
changes are marked with (*) in Figure 2.

| Action | Field label | Description | Format | Constraints |
|---|---|---|---|---|
| | | | | |
| Field added | date_used_lin_age | Indication whether that date was used to construct the linear age model | Text | Selected from pre-defined list: "yes", "no". |
| Field added | date_used_lin_reg | Indication whether that date was used to construct the age model based on linear regression | Text | Selected from pre-defined list: "yes", "no". |
| Field added | date_used_Bchron | Indication whether that date was used to construct the age model based on Bcrhon | Text | Selected from pre-defined list: "yes", "no". |
| Field added | date_used_Bacon | Indication whether that date was used to construct the age model based on Bacon | Text | Selected from pre-defined list: "yes", "no". |
| Field added | date_used_OxCal | Indication whether that date was used to construct the age model based on OxCal | Text | Selected from pre-defined list: "yes", "no". |
| Field added | date_used_copRa | Indication whether that date was used to construct the copRa_based age model | Text | Selected from pre-defined list: "yes", "no". |
| Field added | date_used_StalAge | Indication whether that date was used to construct the age model based on StalAge | Text | Selected from pre-defined list: "yes", "no". |





**Table 3:** Changes made to tables other than the sisal_chronology since the publication of SISALv1
(Atsawawaranunt et al., 2018a; Atsawawaranunt et al., 2018b).

| Table name | Action | Field label | Reason | Format | Constraints |
|---|---|---|---|---|---|
| Dating | Removed "sampling gap" option | *date_type* | This option was never used | Text | Selected from pre-defined list |
| | "others" option changed to "other" | *decay_constant* | Correction of typo | Text | Selected from pre-defined list |
| | Added "other" option | *calib_used* | Option added to accommodate new entities | Text | Selected from pre-defined list |
| | Added "other" option | *date_type* | Option added to accommodate new entities | Text | Selected from pre-defined list |
| Sample | Added "other" option | *original_chronology* | Option added to accommodate new entities | Text | Selected from pre-defined list |
| | Added "other" option | *ann_lam_check* | Option added to accommodate new entities | Text | Selected from pre-defined list |


**Table 4:** Summary of the modifications applied to records already in version 1 (Atsawawaranunt
et al., 2018b) and version 1b (Atsawawaranunt et al., 2019) of the SISAL database. Mistakes in
previous versions of the database were identified as outlined in the Supplementary material and
through analysing the data for the SISAL publications.

| Modification | V1 to v1b | V1b to v2 |
|---|---|---|
| **Site table** | | |
| Number of new sites | 37 | 82 |
| Sites with new entities | 11 | 32 |
| Sites with altered site.site_name altered | 3 | 15 |
| Sites with changes in site.latitude | 4 | 29 |
| Sites with changes in site.longitude | 6 | 32 |
| Sites with changes in site.elevation | 13 | 11 |
| Sites with site.geology updated | 7 | 6 |
| Sites with site.rock_age info updated | 3 | 8 |
| Sites with site.monitoring info updated | 0 | 13 |
| **Entity table** | | |
| Number of new entities | 74 | 236 |
| How many entities were added to pre-existing sites? | 17 | 84 |
| Entities with revised entity_name | 2 | 25 |
| Entities with updated entity.entity_status | 1 | 10 |



| | | |
|---|---|---|
| Entities with altered entity.corresponding current | 0 | 11 |
| Entities with altered entity.depth_ref? | 0 | 1 |
| Entities with altered entity.cover_thickness | 1 | 3 |
| Entities with altered entity.distance_entrance | 0 | 3 |
| Entities with revised entity. speleothem_type | 14 | 4 |
| Entities with revised entity.drip_type | 10 | 2 |
| Entities with altered entity.d13C | 1 | 0 |
| Entities with altered entity.d18O | 1 | 0 |
| Entities with altered entity.d18O_water_equilibrium | 4 | 6 |
| Entities with altered entity.trace_elements | 1 | 2 |
| Entities with altered entity.organics | 1 | 2 |
| Entities with altered entity.fluid_inclusions | 1 | 3 |
| Entities with altered entity.mineralogy_petrology_fabric | 1 | 2 |
| Entities with altered entity.clumped_isotopes | 1 | 3 |
| Entities with altered entity.noble_gas_temperatures | 1 | 2 |
| Entities with altered entity.C14 | 1 | 2 |
| Entities with altered entity.ODL | 1 | 2 |
| Entities with altered entity.Mg_Ca | 1 | 2 |
| Entities with altered entity.contact (mostly correction of typos) | 7 | 32 |
| Entities with altered entity.Data_DOI_URL (revision mostly to permanent links) | 134 | 14 |
| **Dating table** | | |
| Entities with changes in the dating table | 70 | 260 |
| Addition of "Event: hiatus" to an entity | 0 | 3 |
| How many hiatuses had their depth changed? | 2 | 7 |
| Entities with the depths of "Event: start/end of laminations" changed. | 0 | 5 |
| Entities with altered dating.date_type | 11 | 30 |
| Entities with altered dating.depth_dating | 14 | 45 |
| Entities with altered dating.dating_thickness | 14 | 37 |
| Entities with altered dating.material_dated | 5 | 62 |
| Entities with altered dating.min_weight | 13 | 56 |
| Entities with altered dating.max_weight | 19 | 36 |
| Entities with altered dating.uncorr_age | 18 | 48 |
| Entities with altered dating.uncorr_age_uncert_pos | 12 | 53 |
| Entities with altered dating.uncorr_age_uncert_neg | 12 | 41 |
| Entities with altered dating.14C_correction | 17 | 36 |
| Entities with altered dating.calib_used | 13 | 32 |
| Entities with altered dating.date_used | 4 | 51 |
| Entities with altered dating.238U_content | 11 | 45 |
| Entities with altered dating.238U_uncertainty | 16 | 28 |
| Entities with altered dating.232Th_content | 15 | 46 |
| Entities with altered dating.232Th_uncertainty | 14 | 50 |
| Entities with altered dating.230Th_content | 11 | 40 |
| Entities with altered dating.230Th_uncertainty | 15 | 38 |
| Entities with altered dating.230Th_232Th_ratio | 5 | 59 |
| Entities with altered dating.230Th_232Th_ratio_uncertainty | 14 | 48 |
| Entities with altered dating.230Th_238U_activity | 19 | 39 |


| | | |
|---|---|---|
| Entities with altered dating.230Th_238U_activity_uncertainty | 17 | 44 |
| Entities with altered dating.234U_238U_activity | 12 | 51 |
| Entities with altered dating.234U_238U_activity_uncertainty | 11 | 48 |
| Entities with altered dating.ini_230Th_232Th_ratio | 15 | 59 |
| Entities with altered dating.ini_230Th_232Th_ratio_uncertainty | 8 | 60 |
| Entities with altered dating.decay_constant | 17 | 55 |
| Entities with altered dating.corr_age | 17 | 35 |
| Entities with altered dating.corr_age_uncert_pos | 13 | 46 |
| Entities with altered dating.corr_age_uncert_neg | 9 | 47 |
| **Sample table** | | |
| Altered sample.depth_sample | 0 | 15 |
| Altered sample.mineralogy | 0 | 20 |
| Altered sample.arag_corr | 11 | 20 |
| How many entities had their d18O time-series altered (i.e. changes in depth and/or isotope values as in duplicates)? | 13 | 95 |
| How many entities had their d13C time-series altered (i.e. changes in depth and/or isotope values as in duplicates)? | 8 | 64 |
| **Original chronology** | | |
| Entities with altered original_chronology.interp_age | 1 | 42 |
| Entities with altered original_chronology.interp_age_uncert_pos | 0 | 14 |
| Entities with altered original_chronology.interp_age_uncert_neg | 0 | 14 |
| **References** | | |
| How many entities had their references changed (changes/additions/removals)? | 6 | 16 |
| How many citations have a different pub_DOI? | 2 | 16 |
| **Notes** | | |
| Sites with notes removed | 7 | 5 |
| Sites with notes added | 32 | 68 |
| Sites with notes modified | 21 | 34 |

**Table 5**: Information on new speleothem records (entities) added to the SISAL_v2 database from
SISALv1b (Comas-Bru et al., 2019). There may be multiple entities from a single cave, here identified as
the site. Latitude (Lat) and Longitude (Lon) are given in decimal degrees North and East respectively.

| Site_id | Site_name | Lat (N) | Lon (E) | Region | Entity_id | Entity_name | Reference |
|---|---|---|---|---|---|---|---|
| 2 | Kesang cave | 42.87 | 81.75 | China | 620 | CNKS-2 | Cai et al. (2017) |
| | | | | | 621 | CNKS-3 | Cai et al. (2017) |
| | | | | | 622 | CNKS-7 | Cai et al. (2017) |
| | | | | | 623 | CNKS-9 | Cai et al. (2017) |
| 6 | Hulu cave | 32.5 | 119.17 | China | 617 | MSP | Cheng et al. (2006) |
| | | | | | 618 | MSX | Cheng et al. (2006) |



| | | | | | 619 | MSH | Cheng et al. (2006) |
|---|---|---|---|---|---|---|---|
| 12 | Mawmluh cave | 25.2622 | 91.8817 | India | 476 | ML.1 | Kathayat et al. (2018) |
| | | | | | 477 | ML.2 | Kathayat et al. (2018) |
| | | | | | 495 | KM-1 | Huguet et al. (2018) |
| 13 | Ball Gown cave | -17.03 | 125 | Australia | 633 | BGC-5 | Denniston et al. (2013b); Denniston et al. (2017a) |
| | | | | | 634 | BGC-10 | Denniston et al. (2013b); Denniston et al. (2017a) |
| | | | | | 635 | BGC-11_2017 | Denniston et al. (2013b); Denniston et al. (2017a) |
| | | | | | 636 | BGC-16 | Denniston et al. (2013b); Denniston et al. (2017a) |
| 14 | Lehman caves | 39.01 | -114.22 | United States | 641 | CDR3 | Steponaitis et al. (2015) |
| | | | | | 642 | WR11 | Steponaitis et al. (2015) |
| 15 | Baschg cave | 47.2501 | 9.6667 | Austria | 643 | BA-5 | Moseley et al. (2019) |
| | | | | | 644 | BA-7 | Moseley et al. (2019) |
| 23 | Lapa grande cave | -14.37 | -44.28 | Brazil | 614 | LG12B | Stríkis et al. (2018) |
| | | | | | 615 | LG10 | Stríkis et al. (2018) |
| | | | | | 616 | LG25 | Stríkis et al. (2018) |
| 24 | Lapa sem fim cave | -16.1503 | -44.6281 | Brazil | 603 | LSF15 | Stríkis et al. (2018) |
| | | | | | 604 | LSF3_2018 | Stríkis et al. (2018) |
| | | | | | 605 | LSF13 | Stríkis et al. (2018) |
| | | | | | 606 | LSF11 | Stríkis et al. (2018) |
| | | | | | 607 | LSF9 | Stríkis et al. (2018) |



| 27 | Tamboril cave | -16 | -47 | Brazil | 594 | TM6 | Ward et al. (2019) |
|---|---|---|---|---|---|---|---|
| 39 | Dongge cave | 25.2833 | 108.0833 | China | 475 | DA_2009 | Cheng et al. (2009) |
| 54 | Sahiya cave | 30.6 | 77.8667 | India | 478 | SAH-2 | Kathayat et al. (2017) |
| | | | | | 479 | SAH-3 | Kathayat et al. (2017) |
| | | | | | 480 | SAH-6 | Kathayat et al. (2017) |
| 65 | Whiterock cave | 4.15 | 114.86 | Malaysia (Borneo) | 685 | WR12-01 | Carolin et al. (2016) |
| | | | | | 686 | WR12-12 | Carolin et al. (2016) |
| 72 | Ascunsa cave | 45 | 22.6 | Romania | 582 | POM1 | Staubwasser et al. (2018) |
| 82 | Hollywood cave | -41.95 | 171.47 | New Zealand | 673 | HW-1 | Williams et al. (2005) |
| 86 | Modric cave | 44.2568 | 15.5372 | Croatia | 631 | MOD-27 | Rudzka-Phillips et al. (2013) |
| | | | | | 632 | MOD-21 | Rudzka et al. (2012) |
| 105 | Schneckenloch cave | 47.4333 | 9.8667 | Austria | 663 | SCH-6 | Moseley et al. (2019) |
| 113 | Paixao cave | -12.6182 | -41.0184 | Brazil | 611 | PX5 | Strikis et al. (2015) |
| | | | | | 612 | PX7_2018 | Stríkis et al. (2018) |
| 115 | Hölloch im Mahdtal | 47.3781 | 10.1506 | Germany | 664 | HOL-19 | Moseley et al. (2019) |
| 117 | Bunker cave | 51.3675 | 7.6647 | Germany | 596 | Bu2_2018 | Weber et al. (2018) |
| 128 | Buckeye creek | 37.98 | -80.4 | United States | 681 | BCC-9 | Cheng et al. (2019) |
| | | | | | 682 | BCC-10_2019 | Cheng et al. (2019) |
| | | | | | 683 | BCC-30 | Cheng et al. (2019) |
| 135 | Grotte de Piste | 33.95 | -4.246 | Morocco | 464 | GP5 | Ait Brahim et al. (2018) |
| | | | | | 591 | GP2 | Ait Brahim et al. (2018) |
| 138 | Moomi cave | 12.55 | 54.2 | Yemen (Socotra) | 481 | M1-2 | Mangini, Cheng et al., unpublished; Burns et al. (2003) Burns et al. (2004) |
| 140 | Sanbao cave | 31.667 | 110.4333 | China | 482 | SB3 | Wang et al. (2008) |
| | | | | | 483 | SB-10_2008 | Wang et al. (2008) |





| | | | | | 484 | SB11 | Wang et al. (2008) |
|---|---|---|---|---|---|---|---|
| | | | | | 485 | SB22 | Wang et al. (2008) |
| | | | | | 486 | SB23 | Wang et al. (2008) |
| | | | | | 487 | SB24 | Wang et al. (2008) |
| | | | | | 488 | SB25-1 | Wang et al. (2008) |
| | | | | | 489 | SB25-2 | Wang et al. (2008) |
| | | | | | 490 | SB-26_2008 | Wang et al. (2008) |
| | | | | | 491 | SB34 | Wang et al. (2008) |
| | | | | | 492 | SB41 | Wang et al. (2008) |
| | | | | | 493 | SB42 | Wang et al. (2008) |
| | | | | | 494 | TF | Wang et al. (2008) |
| 141 | Sofular cave | 41.4167 | 31.9333 | Turkey | 456 | SO-2 | Badertscher et al. (2011) Fleitmann et al. (2009); Göktürk et al. (2011) |
| | | | | | 687 | SO-4 | Badertscher et al. (2011) |
| | | | | | 688 | SO-6 | Badertscher et al. (2011) |
| | | | | | 689 | SO-14B | Badertscher et al. (2011) |
| 145 | Antro del Corchia | 43.9833 | 10.2167 | Italy | 665 | CC-1_2018 | Tzedakis et al. (2018) |
| | | | | | 666 | CC-5_2018 | Tzedakis et al. (2018) |
| | | | | | 667 | CC-7_2018 | Tzedakis et al. (2018) |
| | | | | | 668 | CC-28_2018 | Tzedakis et al. (2018) |
| | | | | | 669 | CC_stack | Tzedakis et al. (2018) |
| | | | | | 670 | CC27 | Isola et al. (2019) |
| 155 | KNI-51 | -15.3 | 128.62 | Australia | 637 | KNI-51-1 | Denniston et al. (2017a) |
| | | | | | 638 | KNI-51-8 | Denniston et al. (2017a) |
| 160 | Soreq cave | 31.7558 | 35.0226 | Israel | 690 | Soreq-composite185 | Bar-Matthews et al. (2003) |



| 165 | Ruakuri cave | -36.27 | 175.08 | New Zealand | 674 | RK-A | Williams et al. (2010) |
|---|---|---|---|---|---|---|---|
| 165 | Ruakuri cave | -36.27 | 175.08 | New Zealand | 675 | RK-B | Williams et al. (2010) |
| 165 | Ruakuri cave | -36.27 | 175.08 | New Zealand | 676 | RK05-1 | Whittaker (2008) |
| 165 | Ruakuri cave | -36.27 | 175.08 | New Zealand | 677 | RK05-3 | Whittaker (2008) |
| 165 | Ruakuri cave | -36.27 | 175.08 | New Zealand | 678 | RK05-4 | Whittaker (2008) |
| 177 | Santo Tomas cave | 22.55 | -83.84 | Cuba | 608 | CM_2019 | Warken et al. (2019) |
| | | | | | 609 | CMa | Warken et al. (2019) |
| | | | | | 610 | CMb | Warken et al. (2019) |
| 179 | Closani Cave | 45.10 | 22.8 | Romania | 390 | C09-2 | Warken et al. (2018) |
| 182 | Kotumsar cave | 19 | 82 | India | 590 | KOT-I | Band et al. (2018) |
| 192 | El Condor cave | -5.93 | -77.3 | Peru | 592 | ELC-A | Cheng et al. (2013) |
| | | | | | 593 | ELC-B | Cheng et al. (2013) |
| 198 | Lianhua cave, Hunan | 29.48 | 109.5333 | China | 496 | LH-2 | Zhang et al. (2013) |
| 213 | Tausoare cave | 47.4333 | 24.5167 | Romania | 457 | 1152 | Staubwasser et al. (2018) |
| 214 | Cave C126 | -22.1 | 113.9 | Australia | 458 | C126-117 | Denniston et al. (2013a) |
| | | | | | 459 | C126-118 | Denniston et al. (2013a) |
| 215 | Chaara cave | 33.9558 | -4.2461 | Morocco | 460 | Cha2_2018 | Ait Brahim et al. (2018) |
| | | | | | 588 | Cha2_2019 | Ait Brahim et al. (2019) |
| | | | | | 589 | Cha1 | Ait Brahim et al. (2019) |
| 216 | Dark cave | 27.2 | 106.1667 | China | 461 | D1 | Jiang et al. (2013) |
| | | | | | 462 | D2 | Jiang et al. (2013) |
| 217 | E'mei cave | 29.5 | 115.5 | China | 463 | EM1 | Zhang et al. (2018b) |
| 218 | Nuanhe cave | 41.3333 | 124.9167 | China | 465 | NH6 | Wu et al. (2012) |
| | | | | | 466 | NH33 | Wu et al. (2012) |
| 219 | Shennong cave | 28.71 | 117.26 | China | 467 | SN17 | Zhang et al. (2018a) |
| 220 | Baeg-nyong cave | 37.27 | 128.58 | South Korea | 468 | BN-1 | Jo et al. (2017) |





| 221 | La Vierge cave | -19.7572 | 63.3703 | Rodrigues | 469 | LAVI-4 | Li et al. (2018) |
|---|---|---|---|---|---|---|---|
| 222 | Patate cave | -19.7583 | 63.3864 | Rodrigues | 470 | PATA-1 | Li et al. (2018) |
| 223 | Wanxiang cave | 33.32 | 105 | China | 471 | WX42B | Zhang et al. (2008)} |
| | | | | | 679 | WXSM-51 | Johnson et al. (2006) |
| | | | | | 680 | WXSM-52 | Johnson et al. (2006) |
| 224 | Xianglong cave | 33 | 106.33 | China | 472 | XL16 | Tan et al. (2018a) |
| | | | | | 473 | XL2 | Tan et al. (2018a) |
| | | | | | 474 | XL26 | Tan et al. (2018a) |
| 225 | Chiflonkhakha cave | -18.1222 | -65.7739 | Bolivia | 497 | Boto 1 | Apaestegui et al. (2018) |
| | | | | | 498 | Boto 3 | Apaestegui et al. (2018) |
| | | | | | 499 | Boto 7 | Apaestegui et al. (2018) |
| 226 | Cueva del Diamante | -5.73 | -77.5 | Peru | 500 | NAR-C | Cheng et al. (2013) |
| | | | | | 501 | NAR-C-D | Cheng et al. (2013) |
| | | | | | 502 | NAR-C-F | Cheng et al. (2013) |
| | | | | | 503 | NAR-D | Cheng et al. (2013) |
| | | | | | 504 | NAR-F | Cheng et al. (2013) |
| 227 | El Capitan cave | 56.162 | -133.319 | United States | 505 | EC-16-5-F | Wilcox et al. (2019) |
| 228 | Bat cave | 32.1 | -104.26 | United States | 506 | BC-11 | Asmerom et al. (2013) |
| 229 | Actun Tunichil Muknal | 17.1 | -88.85 | Belize | 507 | ATM-7 | Frappier et al. (2002); Frappier et al. (2007); Jamieson et al. (2015) |
| 230 | Marota cave | -12.6227 | -41.0216 | Brazil | 508 | MAG | Stríkis et al. (2018) |
| 231 | Pacupahuain cave | -11.24 | -75.82 | Peru | 509 | P09PH2 | Kanner et al. (2012) |
| 232 | Rio Secreto cave system | 20.59 | -87.13 | Mexico | 510 | Itzamna | Medina-Elizalde et al., (2016); Medina-Elizalde et al. (2017) |



| 233 | Robinson cave | 33 | -107.7 | United States | 511 | KR1 | Polyak et al. (2017) |
|---|---|---|---|---|---|---|---|
| 234 | Santana cave | -24.5308 | -48.7267 | Brazil | 512 | St8-a | Cruz et al. (2006) |
| | | | | | 513 | St8-b | Cruz et al. (2006) |
| 235 | Cueva del Tigre Perdido | -5.9406 | -77.3081 | Peru | 514 | NC-A | van Breukelen et al. (2008) |
| | | | | | 515 | NC-B | van Breukelen et al. (2008) |
| 236 | Toca da Boa Vista | -10.1602 | -40.8605 | Brazil | 516 | TBV40 | Wendt et al. (2019) |
| | | | | | 517 | TBV63 | Wendt et al. (2019) |
| 237 | Umajalanta cave | -18.12 | -65.77 | Bolivia | 518 | Boto 10 | Apaestegui et al. (2018) |
| 238 | Akalagavi cave | 14.9833 | 74.5167 | India | 519 | MGY | Yadava et al. (2004) |
| 239 | Baluk cave | 42.433 | 84.733 | China | 520 | BLK12B | Liu et al. (2019) |
| 240 | Baratang cave | 12.0833 | 92.75 | India | 521 | AN4 | Laskar et al. (2013) |
| | | | | | 522 | AN8 | Laskar et al. (2013) |
| 241 | Gempa bumi cave | -5 | 120 | Indonesia (Sulawesi) | 523 | GB09-03 | Krause et al. (2019) |
| | | | | | 524 | GB11-09 | Krause et al. (2019) |
| 242 | Haozhu cave | 30.6833 | 109.9833 | China | 525 | HZZ-11 | Zhang et al. (2016) |
| | | | | | 526 | HZZ-27 | Zhang et al. (2016) |
| 243 | Kailash cave | 18.8445 | 81.9915 | India | 527 | KG-6 | Gautam et al. (2019) |
| 244 | Lianhua cave, Shanxi | 38.1667 | 113.7167 | China | 528 | LH1 | Dong et al. (2018) |
| | | | | | 529 | LH4 | Dong et al. (2018) |
| | | | | | 530 | LH5 | Dong et al. (2018) |
| | | | | | 531 | LH6 | Dong et al. (2018) |
| | | | | | 532 | LH9 | Dong et al. (2018) |
| | | | | | 533 | LH30 | Dong et al. (2018) |
| 245 | Nakarallu cave | 14.52 | 77.99 | India | 534 | NK-1305 | Sinha et al. (2018) |
| 246 | Palawan cave | 10.2 | 118.9 | Malaysia (Northern Borneo) | 535 | SR02 | Partin et al. (2015) |
| 247 | Shalaii cave | 35.1469 | 45.2958 | Iraq | 536 | SHC-01 | Marsh et al. (2018); Amin Al- |





| | | | | | | | | Manmi et al. (2019) |
|---|---|---|---|---|---|---|---|---|
| | | | | | 537 | SHC-02 | | Marsh et al. (2018); Amin Al-Manmi et al. (2019) |
| 248 | Shenqi cave | 28.333 | 103.1 | China | 538 | SQ1 | | Tan et al. (2018b) |
| | | | | | 539 | SQ7 | | Tan et al. (2018b) |
| 249 | Shigao cave | 28.183 | 107.167 | China | 540 | SG1 | | Jiang et al. (2012) |
| | | | | | 541 | SG2 | | Jiang et al. (2012) |
| 250 | Wuya cave | 33.82 | 105.43 | China | 542 | WY27 | | Tan et al. (2015) |
| | | | | | 543 | WY33 | | Tan et al. (2015) |
| 251 | Zhenzhu cave | 38.25 | 113.7 | China | 544 | ZZ12 | | Yin et al. (2017) |
| 252 | Andriamaniloke | -24.051 | 43.7569 | Madagascar | 545 | AD4 | | Scroxton et al. (2019) |
| 253 | Hoq cave | 12.5866 | 54.3543 | Yemen (Socotra) | 546 | Hq-1 | | Van Rampelbergh et al. (2013) |
| | | | | | 547 | STM1 | | Van Rampelbergh et al. (2013) |
| | | | | | 548 | STM6 | | Van Rampelbergh et al. (2013) |
| 254 | PP29 | -34.2078 | 22.0876 | South Africa | 549 | 46745 | | Braun et al. (2019b) |
| | | | | | 550 | 46746-a | | Braun et al. (2019b) |
| | | | | | 551 | 46747 | | Braun et al. (2019b) |
| | | | | | 552 | 138862.1 | | Braun et al. (2019b) |
| | | | | | 553 | 138862.2a | | Braun et al. (2019b) |
| | | | | | 554 | 142828 | | Braun et al. (2019b) |
| | | | | | 555 | 46746-b | | Braun et al. (2019b) |
| | | | | | 556 | 138862.2b | | Braun et al. (2019b) |
| 255 | Mitoho | -24.0477 | 43.7533 | Madagascar | 557 | MT1 | | Scroxton et al. (2019) |

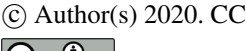


| 256 | Lithophagus cave | 46.828 | 22.6 | Romania | 558 | LFG-2 | Lauritzen and Onac (1999) |
|---|---|---|---|---|---|---|---|
| 257 | Akcakale cave | 40.4498 | 39.5365 | Turkey | 559 | 2p | Jex et al. (2010); Jex et al. (2011); Jex et al. (2013) |
| 258 | B7 cave | 49 | 7 | Germany | 560 | STAL-B7-7 | Niggemann et al. (2003b) |
| 259 | Cobre cave | 42.98 | -4.37 | Spain | 561 | PA-8 | Osete et al. (2012); Rossi et al. (2014) |
| 260 | Crovassa Azzurra | 39.28 | 8.48 | Italy | 562 | CA | Columbu et al. (2019) |
| 261 | El Soplao cave | 43.2962 | -4.3937 | Spain | 563 | SIR-1 | Rossi et al. (2018) |
| 262 | Bleßberg cave | 50.4244 | 11.0203 | Germany | 564 | BB-1 | Breitenbach et al. (2019) |
| | | | | | 565 | BB-3 | Breitenbach et al. (2019) |
| 263 | Orlova Chuka cave | 43.5937 | 25.9597 | Bulgaria | 566 | ocz-6 | Pawlak et al. (2019) |
| 264 | Strašna peć cave | 44.0049 | 15.0388 | Croatia | 567 | SPD-1 | Lončar et al. (2019) |
| | | | | | 568 | SPD-2 | Lončar et al. (2019) |
| 265 | Coves de Campanet | 39.7937 | 2.9683 | Spain | 569 | CAM-1 | Dumitru et al. (2018) |
| 266 | Cueva Victoria | 37.6322 | -0.8215 | Spain | 570 | Vic-III-4 | Budsky et al. (2019) |
| 267 | Gruta do Casal da Lebre | 39.3 | -9.2667 | Portugal | 571 | GCL6 | Denniston et al. (2017b) |
| 268 | Pere Noel cave | 50 | 5.2 | Belgium | 572 | PN-95-5 | Verheyden et al. (2000); Verheyden et al. (2014) |
| 269 | Gejkar cave | 35.8 | 45.1645 | Iraq | 573 | Gej-1 | Flohr et al. (2017) |
| 270 | Gol-E-Zard cave | 35.84 | 52 | Iran | 574 | GZ14-1 | Carolin et al. (2019) |
| 271 | Jersey cave | -35.72 | 148.49 | Australia | 575 | YB-F1 | Webb et al. (2014) |
| 272 | Metro cave | -41.93 | 171.47 | New Zealand | 576 | M-1 | Logan (2011) |
| 273 | Crystal cave | 36.59 | -118.82 | United States | 577 | CRC-3 | McCabe-Glynn et al. (2013) |
| 274 | Terciopelo cave | 10.17 | -85.33 | Costa Rica | 578 | CT-1 | Lachniet et al. (2009) |
| | | | | | 579 | CT-5 | Lachniet et al. (2009) |





| | | | | | 580 | CT-6 | Lachniet et al. (2009) |
|---|---|---|---|---|---|---|---|
| | | | | | 581 | CT-7 | Lachniet et al. (2009) |
| 275 | Buraca Gloriosa | 39.5333 | -8.7833 | Portugal | 583 | BG41 | Denniston et al. (2017b) |
| | | | | | 584 | BG66 | Denniston et al. (2017b) |
| | | | | | 585 | BG67 | Denniston et al. (2017b) |
| | | | | | 586 | BG611 | Denniston et al. (2017b) |
| | | | | | 587 | BG6LR | Denniston et al. (2017b) |
| 276 | Béke cave | 48.4833 | 20.5167 | Hungary | 595 | BNT-2 | Demény et al. (2019) |
| | | | | | | | Czuppon et al. (2018) |
| 277 | Huagapo cave | -11.27 | -75.79 | Peru | 597 | P00-H2 | Kanner et al. (2013) |
| | | | | | 598 | P00-H1 | Kanner et al. (2013) |
| | | | | | 599 | P09-H1b | Burns et al. (2019) |
| | | | | | 600 | P10-H5 | Burns et al. (2019) |
| | | | | | 601 | P10-H2 | Burns et al. (2019) |
| | | | | | 602 | PeruMIS6Composite | Burns et al. (2019) |
| 278 | Pink Panther cave | 32 | -105.2 | United States | 613 | PP1 | Asmerom et al. (2007) |
| 279 | Staircase cave | -34.2071 | 22.0899 | South Africa | 624 | 46322 | Braun et al. (2019b) |
| | | | | | 625 | 46330-a | Braun et al. (2019b) |
| | | | | | 626 | 46861 | Braun et al. (2019b) |
| | | | | | 627 | 50100 | Braun et al. (2019b) |
| | | | | | 628 | 142819 | Braun et al. (2019b) |
| | | | | | 629 | 142820 | Braun et al. (2019b) |
| | | | | | 630 | 46330-b | Braun et al. (2019b) |
| 280 | Atta cave | 51.1 | 7.9 | Germany | 639 | AH-1 | Niggemann et al. (2003a) |
| 281 | Venado cave | 10.55 | -84.77 | Costa Rica | 640 | V1 | Lachniet et al. (2004) |



| 282 | Wadi Sannur cave | 28.6167 | 31.2833 | Eqypt | 691 | WS-5d | El-Shenawy et al. (2018) |
|---|---|---|---|---|---|---|---|
| 283 | Babylon cave | -41.95 | 171.47 | New Zealand | 645 | BN-1 | Williams et al. (2005) |
| | | | | | 646 | BN-2 | Williams et al. (2005) |
| | | | | | 647 | BN-3 | P. Williams et al., unpublished |
| 284 | Creighton`s cave | -40.63 | 172.47 | New Zealand | 648 | CN-1 | Williams et al. (2005) |
| 285 | Disbelief cave | -38.82 | 177.52 | New Zealand | 649 | Disbelief | Lorrey et al. (2008) |
| 286 | La Garma cave | 43.4306 | -3.6658 | Spain | 650 | GAR-01_drill | Baldini et al. (2015); Baldini et al. (2019) |
| | | | | | 651 | GAR-01_laser_d18O | Baldini et al. (2015) |
| | | | | | 652 | GAR-01_laser_d13C | Baldini et al. (2015) |
| 287 | Twin Forks cave | -40.63 | 172.48 | New Zealand | 653 | TF-2 | Williams et al. (2005) |
| 288 | Wet Neck cave | -40.7 | 172.48 | New Zealand | 654 | WN-4 | Williams et al. (2005) |
| | | | | | 655 | WN-11 | Williams et al. (2005) |
| 289 | Gassel Tropfsteinhöhle | 47.8228 | 13.8428 | Austria | 656 | GAS-12 | Moseley et al. (2019) |
| | | | | | 657 | GAS-13 | Moseley et al. (2019) |
| | | | | | 658 | GAS-22 | Moseley et al. (2019) |
| | | | | | 659 | GAS-25 | Moseley et al. (2019) |
| | | | | | 660 | GAS-27 | Moseley et al. (2019) |
| | | | | | 661 | GAS-29 | Moseley et al. (2019) |
| 290 | Grete-Ruth Shaft | 47.5429 | 12.0272 | Austria | 662 | HUN-14 | Moseley et al. (2019) |
| 292 | Limnon cave | 37.9605 | 22.1403 | Greece | 671 | KTR-2 | Peckover et al. (2019) |
| 293 | Tham Doun Mai | 20.75 | 102.65 | Laos | 672 | TM-17 | Wang et al. (2019) |
| 294 | Palco cave | 18.35 | -66.5 | Puerto Rico | 684 | PA-2b | Rivera-Collazo et al. (2015) |
| 179 | Closani Cave | 45.10 | 22.8 | Romania | 390 | C09-2 | Warken et al. (2018) |




**Table 6:** Percentage of entities uploaded to the different versions of the SISAL database with
respect to the number of records identified by the SISAL working group as of November 2019.
The number of identified records includes potentially superseded speleothem records. Regions
are defined as: Oceania (-60° < Lat < 0°; 90° < Lon < 180°); Asia (0° < Lat < 60°; 60° < Lon < 130°);
Middle East (7.6° < Lat < 50°; 26° < Lon < 59°); Africa (-45° < Lat < 36.1°; -30° < Lon < 60°; with
records in the Middle East region removed); Europe (36.7° < Lat < 75°; -30° < Lon < 30°; plus
Gibraltar and Siberian sites); South America (S. Am; -60° < Lat < 8°; -150° < Lon < -30°); North and
Central America (N./C. Am; 8.1° < Lat < 60°; -150° < Lon < -50°)

| Region | Version 1 | | Version 1b | | Version 2 | |
|---|---|---|---|---|---|---|
| | Entities | Sites | Entities | Sites | Entities | Sites |
| **Oceania** | 47.7 | 36.7 | 56.8 | 51.0 | 80.2 | 69.4 |
| **Asia** | 36.2 | 28.8 | 41.1 | 33.3 | 64.8 | 48.5 |
| **Middle East** | 21.2 | 31.1 | 28.8 | 35.6 | 42.3 | 48.9 |
| **Africa** | 63.2 | 62.5 | 63.2 | 62.5 | 73.7 | 87.5 |
| **Europe** | 48.0 | 51.9 | 54.6 | 58.7 | 75.3 | 77.9 |
| **S. Am** | 30.6 | 39.5 | 40.8 | 50.0 | 77.6 | 73.7 |
| **N./C. Am** | 35.7 | 36.7 | 51.8 | 56.7 | 70.5 | 73.3 |




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
