# Peer review of "SISALv2: A comprehensive speleothem isotope database with multiple age-depth models"

_Earth System Science Data, 2020_

## Referee Comment (RC1) · Oliver Bothe (Referee) · 22 Apr 2020

Dear editor, dear authors,

Comas-Bru et al. present an update to the database first presented in Atsawawaranunt et al. (2018) of the Speleothem Isotope Synthesis and Analysis (SISAL) working group. The update consists of additional records, the extension of other records, changes, corrections and clarifications in the database, and, most importantly, the creation of new chronologies for age-depth modelling. The authors used seven different methods for obtaining chronologies.

[Figure]

By collecting the data in one place and by providing new and revised chronologies for the data, this manuscript and the dataset is a welcome and valuable addition to our sources of information about the three core intervals of earth's recent past, the Common Era of the last 2000 years, the period since the last glacial maximum, and the last interglacial.

I do not have any major comments and even minor comments are rare. However, this may be in part due to the fact that I am not a specialist for speleothems.

There is one suggestion for the future of the database. I would like to invite the authors to reconsider the general structure of the database and/or to provide a set of functions that allow an even easier interaction with the data than is already the case. Obviously, it is unlikely that there is a set of tools that accounts for all preferences of all potential users, and neither is it likely that there is a database structure that is intuitive even for the most inexperienced user. This suggestion originates from my struggles to access the data.

Minor:

Suggestion: Add a list of abbreviations and technical terms. From my point of view, a database and a database paper is of interest for a community that extends beyond the specialists in a field. Therefore it might be advisable to provide a table or an appendix with definitions of technical terms and abbreviations. Otherwise, already Figure 1 may overwhelm some potential users.

Line 16: The authors write here 294 and later 293. Obviously "cave sites" and "cave systems" may mean different things, but I would like to ask the authors to clarify.

Line 81: 10.5281/zenodo.3591197 is not a valid doi, i.e. the link is broken. This occurs again at line 271. The doi in line 270 is not valid either. Finally, in section 6 (lines 273, 274, 276) and in the supplementary materials there are a few dois given as links but missing the "doi.org"-part and therefore not working directly.

Line 96: "This is consistent" implies that there are differences to these previous approaches. If these differences can be summarised shortly, it may be helpful to detail them.

Provide a script to reproduce Figure 5. I would like to suggest to provide a script that reproduces Figure 5. Parts of this are already included in the GitHub-repository for SISAL.AM, cf. the plot_sisal_overview.R script. However, the final.plot function is only partially doing this and the script does not run as it is. The reading of the input-files has two errors (that can be easily corrected).

Technicalities:

Some of the following technicalities are purely subjective. Some are meant to capture already at this stage some of the annotations that will come because of Copernicus' copy editing efforts.

Line 22: Please rephrase the sentence in combination with the parentheses. I am also unsure whether the tense of the verb in the parentheses is correct.

Line 23: I stumbled between the first and second sentence of the paragraph as I felt they were rather badly connected. But that is rather subjective.

Line 25: Again subjectively, I felt there should already be a new paragraph here.

Line 40: I assume it should read "pointed to the" instead of "pointed the".

Line 48: The authors use two different notations for COPRA; COPRA and copRa. It may be that copRa is meant only to refer to the implementation of COPRA, but that does not become clear. I invite the authors to either only use one notation or to clearly specify that copRa is COPRA for R.

In this context, I also would invite the authors of copRa to make copRa publically available. Giving an official acronym to the author's COPRA implementation in R already suggests such an availability.

Similarly it would be nice, if the authors' updated version of StalAge could be made available. This may be already the case, then I missed the pointer.

It may be that all what I am writing here is already fulfilled by Roesch (2020, https://github.com/paleovar/SISAL.AM). If that is the case then please put the link at all positions where it is relevant. Oh, and the link to github.com/palaeovar/SISAL.AM given by the authors doesn't work.

Line 154: The authors mention the function pchip but not the package. I ask them to add the package and a citation for it. There may be other instances where this is necessary.

Personal communications: If I remember correctly, Copernicus' editors are asking authors to provide the year of personal communication and also full names.

Similarly, the copy editors/typesetting editors are going to check for completeness of citations. I think I noted some missing information, e.g., the DOI for Rehfeld and Kurths (2014). Maybe the authors want to check all citations for completeness already at this stage.

All links: please provide last accessed dates for all URLs.

R packages: I did not check in detail - and I do not always follow this idea myself - but I think it would be nice if the authors could not only mention the R packages they use but also provide references for each of them. It may be that they already do this. I only noted that, e.g., for rBacon there is not a direct reference given, but maybe this is already fulfilled with the references in line 124. Another example is the mentioning of Hmisc on line 99.

Software in general: The last comment, obviously, also applies to all other sorts of software, e.g., github-links in the manuscript. Similarly, the authors probably should also provide references for the zenodo-dois in the manuscript.

---

## Author Comment (AC1) · 8 May 2020

Dear Oliver Bothe,
Thanks for your positive comments and suggestions. Please, find our answers below.

**General comments**

There is one suggestion for the future of the database. I would like to invite the authors to reconsider the general structure of the database and/or to provide a set of functions that allow an even easier interaction with the data than is already the case. Obviously,

it is unlikely that there is a set of tools that accounts for all preferences of all potential users, and neither is it likely that there is a database structure that is intuitive even for the most inexperienced user. This suggestion originates from my struggles to access the data.

Accessing and querying any database is challenging for non-specialists but, as the reviewer acknowledges, it would be impossible to provide functions for all types of software and that addresses all the preferences that data users may have. To facilitate potential database users, we have created a set of "example queries" that are available along with the database files. While these 27 queries do not cover all potential needs of the researchers using the database, they provide a starting point. The existence of the example queries is mentioned in the README file available with the database in the data repository, but we will add text in the manuscript to make this clearer.

The structure of the database was agreed with the wider speleothem community as the most efficient way to capture the metadata and data available from speleothem records. A detailed description of the database and all its parameters is available in the paper describing the first version of the database (Atsawawaranunt et al., 2018). We will make sure that this is clear in the text. Note that it is also possible to download the individual tables comprising the database, allowing a user to reconfigure this in any format they prefer.

**Minor**

**Suggestion**: Add a list of abbreviations and technical terms. From my point of view, a database and a database paper is of interest for a community that extends beyond the specialists in a field. Therefore it might be advisable to provide a table or an appendix with definitions of technical terms and abbreviations. Otherwise, already Figure 1 may overwhelm some potential users.

In this paper, we did not define all concepts presented in the current database structure

because these are already defined in Atsawawaranunt et al., 2018. This publication is cited in the manuscript as the source to look at for a detailed description of the database (L170-172: "*The structure and contents of all tables except the new sisal_chronology table are described in detail in Atsawawaranunt et al. (2018a). Here, we focus on the new sisal_chronology table and on the changes that were made to other tables in order to accommodate this new table (See section 2.3).*").

We are unsure as to whether providing that information again in the supplementary material of this paper would make things substantially different than referring to a paper previously published in the same journal. However, we will make it clear earlier in the manuscript that all concepts that appear in the database are defined in that publication and we hope that this will address the reviewer's concern satisfactorily.

Line 16: The authors write here 294 and later 293. Obviously "cave sites" and "cave systems" may mean different things, but I would like to ask the authors to clarify.

We used cave sites and cave systems interchangeably but will change it to cave sites throughout.

Line 81: 10.5281/zenodo.3591197 is not a valid doi, i.e. the link is broken.

Thank you for spotting this. The link to the ensemble file is now working.

This occurs again at line 271.

Thank you for spotting this. This link is now working properly.

The doi in line 270 is not valid either.

Thank you for spotting this. This link is now working properly.

Finally, in section 6 (lines 273,274, 276) and in the supplementary materials there are a few dois given as links but missing the "doi.org"-part and therefore not working directly.

Thanks. We will revise the hyperlinks in the manuscript as well as in the supplementary material before submitting a revised version of this manuscript.

Line 96:"This is consistent" implies that there are differences to these previous approaches. If these differences can be summarised shortly, it may be helpful to detail them.

By "this is consistent" we mean that our approach is equivalent, in that it produces highly similar results (apart from small numerical differences we expect due to the choice of random number generator and interpolation depths as well as, for COPRA the different programming language). We will clarify this in the revised document.

Provide a script to reproduce Figure 5. I would like to suggest to provide a script that reproduces Figure 5. Parts of this are already included in the GitHubrepository for SISAL.AM, cf. the plot_sisal_overview.R script. However, the final.plot function is only partially doing this and the script does not run as it is. The reading of the input-files has two errors (that can be easily corrected).

We have updated the GitHub repository for SISAL.AM with a script to successfully produce figure 5 as requested by the reviewer: https://github.com/paleovar/SISAL.AM/ tree/master/SISAL_plot_functions

**Technicalities**

Some of the following technicalities are purely subjective. Some are meant to capture already at this stage some of the annotations that will come because of Copernicus' copy editing efforts.

Line 22: Please rephrase the sentence in combination with the parentheses. I am also unsure whether the tense of the verb in the parentheses is correct.

We will rephrase this sentence to: "Speleothems are a rich terrestrial palaeoclimate archive that forms from infiltrating rainwater after it percolates through the soil, epikarst, and carbonate bedrock."

Line 23: I stumbled between the first and second sentence of the paragraph as I felt

they were rather badly connected. But that is rather subjective.

We agree with this comment and we will revise these two sentences to: "Speleothems are a rich terrestrial palaeoclimate archive that forms from infiltrating rainwater after it percolates through the soil, epikarst, and carbonate bedrock. In particular, stable oxygen and carbon isotope ($\delta$18O, $\delta$13C) measurements made on speleothems have been widely used to reconstruct regional and local hydroclimate changes."

Line 25: Again subjectively, I felt there should already be a new paragraph here.

We agree with the reviewer and we will move the sentence starting with "The Speleothem Isotope Synthesis..." to a new paragraph.

Line 40: I assume it should read "pointed to the" instead of "pointed the".

We agree and we will apply this change.

Line 48: The authors use two different notations for COPRA; COPRA and copRa. It may be that copRa is meant only to refer to the implementation of COPRA, but that does not become clear. I invite the authors to either only use one notation or to clearly specify that copRa is COPRA for R. In this context, I also would invite the authors of copRa to make copRa publically available. Giving an official acronym to the author's COPRA implementation in R already suggests such an availability.

Indeed, copRa is the R implementation of COPRA created specifically for this paper. We will revise the text to make this clear.
As stated in the data availability section, this new copRa implementation is already publicly available at https://github.com/paleovar/SISAL.AM.

Similarly it would be nice, if the authors' updated version of StalAge could be made available. This may be already the case, then I missed the pointer. It may be that all what I am writing here is already fulfilled by Roesch (2020, https://github.com/paleovar/SISAL.AM). If that is the case then please put the link at all positions where it is relevant.

[Figure]

Yes, as stated in the data availability section, the modified version of StalAge is already publicly available at https://github.com/paleovar/SISAL.AM. We will add the links were appropriate in addition to the "data availability section" to make this clear.

Oh, and the link to github.com/palaeovar/SISAL.AM given by the authors does not work.

Apologies, the correct link is https://github.com/paleovar/SISAL.AM (without the "a" of palaeo). We will correct the link in the manuscript.

Line 154: The authors mention the function *pchip* but not the package. I ask them to add the package and a citation for it. There may be other instances where this is necessary.

The *pchip* function comes from the signal R package, we will add this in the reference list.
We will provide the package name and a citation for it every time that an R function is mentioned in the manuscript.

**Personal communications:** If I remember correctly, Copernicus' editors are asking authors to provide the year of personal communication and also full names.

We will revise this to: "(Christoph Bronk Ramsey, personal communication, 2019)."

Similarly, the copy editors/typesetting editors are going to check for completeness of citations. I think I noted some missing information, e.g., the DOI for Rehfeld and Kurths (2014). Maybe the authors want to check all citations for completeness already at this stage.

Thanks for spotting this. We will revise all DOIs and references before submitting a revised version of the manuscript.

All links: please provide last accessed dates for all URLs.

We will do.

<... >

**R packages:** I did not check in detail - and I do not always follow this idea myself - but I think it would be nice if the authors could not only mention the R packages they use but also provide references for each of them. It may be that they already do this. I only noted that, e.g., for rBacon there is not a direct reference given, but maybe this is already fulfilled with the references in line 124. Another example is the mentioning of Hmisc on line 99.

We will make sure that all R packages that we used are cited were appropriate and that links are always copied where relevant.

There is an R package with rBacon. We mention it in the data availability section: "rBacon package (version 2.3.9.1) is available on CRAN (https://cran.r-project.org/web/packages/rbacon/index.html)." but we will also add its citation where appropriate in the text.

We will also cite the packages of Hmisc (L99) and pchip (L154) in the manuscript.

**Software in general:** The last comment, obviously, also applies to all other sorts of software, e.g., github-links in the manuscript. Similarly, the authors probably should also provide references for the zenodo-dois in the manuscript.

We will double check that all links are properly working before submitting a revised version of the manuscript. We will also properly cite the zenodo-dois mentioned in the manuscript.

---

## Referee Comment (RC2) · J. W. Partin (Referee) · 24 Jun 2020

Paleoclimate data compilations are unsung heroes right now for paleoclimate and climate science. Well, not so much unsung, as under funded. They are huge volunteer efforts that will lead to some interesting new science based on the abundance of machine readable data, i.e. Big Data. For this aspect alone, I congratulate and thank the team, and I recommend to publish this manuscript. I was also happy to see that SISAL V2 is upping the game to include alternate age modeling techniques. A major strength of speleothem records are absolute ages via the U-Th decay series. However, I think that this paper has the opportunity, given the long list of top notch authors, to

establish some firm guidelines for future papers using speleothems to reconstruct past climate. And since they wrote the code, and are the experts on V2, I suggest a few more calculations are warranted for this publication to aid in the use of the database.

Can you please describe in more detail how the 95% confidence intervals are calculated (line 227) in the SISAL chronologies? Lines 204-210 are hard to follow. I'm not sure, but I think it is the 95% spread in the ages using all of the age models, i.e. the spread in the curves in Figure 5 a and b. If so, I wholeheartedly support this idea. If not, then please describe in more detail.

I recommend that the authors take this opportunity to strengthen their language for section 4, and provide a clear road map for exactly how future users should use their code on new speleothem records to produce not only alternate MC age models from a single technique, but alternate age models from different techniques as well – as done here. I acknowledge that not all techniques are possible for all records. But still - all techniques worked on 503 or 504 of 533 records (Figure 4). So in the future, people should utilize as many as possible.

Going on the premise that I'm not sure of the methodology, I advocate that the 95% spread in all of the Monte Carlo ensembles from all of the age modeling techniques that are successfully executed is used for the final age model (Figure 5a, 5b). Or at the very least, the spread in the medians of the different age models, though an MC ensemble is quite useful. The idea here is simple: we work so hard to get these absolute ages that we should be rewarded for getting multiple ages on a sample, not penalized. Errors in age models should not be constrained simply by the analytical (or analytical plus correction) based error bars.

For example in Figure 5, there are a string of ages from 3400 – 3550 year BP that all follow each other. For this region of the d18O curve, there is fairly good agreement between the various age modeling techniques. Therefore, when all of those ages are viewed together, our confidence in the timing of any d18O excursion is less than that

based on the error bars on each individual date (seen by less 'blur' in Figure 5c between the alternate age models). In other words, the multiple ages help to decrease our uncertainty to less than that of the analytical error bars on each U-Th dates. It's a bit like decreasing the signal to noise ratio by taking more measurements (by the square root of N). (Again SISAL may be doing this, but I'm not sure)

To quantify the degree to which multiple age modeling techniques may reduce temporal uncertainty, I recommend that this manuscript includes a plot of the average of the analytical error in a record versus the average SISAL 95% chronology error in a record (i.e. average of 5b). Is the fit to that scatter plot a 1:1 line? Or is there a systematic reduction in the error across many records in the database b/c of time periods like 3400-3550 BP in Figure 5? Or do problematic areas, like unresolved hiatuses, compensate the reduction in errors for when the age model is tightly constrained? This would be an enlightening plot.

Please give more detail in the text of the principles used in your calculations for when the SISAL chronology decides that there is a hiatus in the record. While you reference Breitenbach, 2012, it would be good to review the guiding principles that SISAL is using in lines 83-86 in more detail to make the manuscript more self contained. Also, what happens if there is disagreement among the various techniques about a hiatus - how does SISAL decide on a 'yes' or 'no' to split the record? Does majority rule??
* * *

---

## Author Comment (AC2) · 5 Aug 2020

[a4paper,12pt]article xcolor parskip  url

Dear Jud W. Partin,

Thanks for your comments and suggestions.  Indeed, data compilations are an enormous community-based effort that we believe will have a strong impact on how science is done. Please, find our answers below highlighted in blue.

[Figure]

Can you please describe in more detail how the 95% confidence intervals are calculated (line 227) in the SISAL chronologies? I'm not sure, but I think it is the 95% spread in the ages using all of the age models, i.e. the spread in the curves in Figure 5 a and b. If so, I wholeheartedly support this idea. If not, then please describe in more detail.

The 95% confidence intervals are the spread for each type of age model separately not the spread considering all of the age models. Specifically, the SISAL chronology table has three columns per age-depth model technique. The first is the median age-depth model and the other two are their corresponding 2 sigma confidence intervals. These confidence intervals have been calculated from the spread of the individual ensembles for each technique that support this approach (specifically, linear interpolation, linear regression, Bchron, Bacon, OxCal and copRa)) and the last 1000 have been kept – and made available – for further analyses. For StalAge we report the uncertainties of the returned age model which are internally calculated and based on iterative fits as well as dating uncertainty. Fig. 5a and 5b show the median age models for each technique (the mean for StalAge). Each of these lines has corresponding confidence intervals which are individually reported.

We have not attempted to merge all SISAL chronologies for any given entity. Different modelling approaches give stronger weighting to some age determinations, so averaging across several methods could yield age-depth relationships that are unrealistic and/or not robust compared to the available U-Th dates.

We will add a statement on how the uncertainties are obtained in the description of each age-depth model technique in section 2.1.

Lines 204-210 are hard to follow. (Author's comment: Please note that we have separated this comment from the paragraph above as it refers to different things).

*Original text in L204-210 copied for reference: "The conception and the test of the R workflow, integrating all methods but OxCal, was outlined in Roesch and Rehfeld*

*(2019) and includes automatized checks for the final chronologies except for OxCal. The quality control parameters obtained from OxCal were compared with the recommended values of Agreement Index (A) larger than 60 % and Convergence (C) larger than 95 %, in accordance with the guidelines in Bronk Ramsey (2008). In addition to both model agreement and P-Sequence convergence meeting these criteria, at least 90 % of individual dates had to have an acceptable Agreement and Convergence themselves. OxCal age-depth models failing to meet these criteria were not included in the SISAL chronology table."*

We acknowledge that this paragraph needs rewriting and we will add the following text instead:

*"We used an automated approach to age-depth modelling in R because of the large number of records. Roesch and Rehfeld (2019) have described the basic workflow concept and tested it using all of the age-modelling approaches used here except OxCal. The basic workflow involves step-by-step inspection and formatting of the data for the different methods, and the use of pre-defined parameter choices specific to each method. Each age-modelling method is called sequentially. An error message is recorded in the log file if a particular age-modelling method fails, and the algorithm then progresses to the next method. If output is produced for a particular age-modelling method, these age models are checked for monotonicity. Finally, the output standardization routine writes out, for each entity and age-modelling approach, the median age model, the ensembles (if applicable) and information of which hiatuses and dates were used in the construction of the age models. These outputs are then added to the sisal-chronology table (Table 2). All functions are available at* https:// github.com/ paleovar/ SISAL.AM *(last access: 23 July 2020) and CRAN (*https:// cran.r-project.org/ web/ packages/ rbacon/ index.html*; last access: 31 January 2020).*

*The general approach for the OxCal age models was similar, and step-by-step details and scripts are provided in* https:// doi.org/ 10.5281/ zenodo.3586280 *(Amirnezhad-*

*Mozhdehi and Comas-Bru, 2019). The quality control parameters obtained from OxCal were compared with the recommended values of Agreement Index (A) larger than 60% and Convergence (C) larger than 95%, in accordance with the guidelines in Bronk Ramsey (2008), both for the overall model and for at least 90% of the individual dates. OxCal age-depth models failing to meet these criteria were not included in the sisal-chronology table (Table 2)."*

We agree that the spread of the median age models is useful. However, we do not think it is useful to calculate cross-model uncertainties based on the medians/ensembles of all age models. As explained above, in many cases the resulting merged chronology would not be consistent with the available U-Th dates. However, we supply all the data (medians, uncertainties, ensembles) so that this could be done by individual researchers on an entity by entity basis. Medians and uncertainties are available in the database and ensembles are stored in zenodo: http://doi.org/10.5281/zenodo.3816804 (Rehfeld et al., 2020).

Our approach does not penalize records with many dates. Usually, the better the spread in dates along a speleothem sample, the more constrained the age-depth model, and the lower the final uncertainty of the best-estimate median age model. However, it should be noted that this is not the case when there is a large number of conflicting U-Th dates with high analytical and correction uncertainties as the resulting age-depth model realisations will substantially differ amongst them.

For example, in Figure 5, there are a string of ages from 3400 – 3550 year BP that all follow each other. For this region of the d18O curve, there is fairly good agreement between the various age modeling techniques. Therefore, when all of those ages are viewed together, our confidence in the timing of any d18O excursion is less than that based on the error bars on each individual date (seen by less 'blur' in Figure 5c between the alternate age models). In other words, the multiple ages help to decrease our uncertainty to less than that of the analytical error bars on each U-Th dates. It's a bit

like decreasing the signal to noise ratio by taking more measurements (by the square root of N). (Again SISAL may be doing this, but I'm not sure)

We agree with the reviewer that taking into account multiple chronologies for a given record provides more insights into the age-depth relationship. However, given that all the age models are consistent with the available dates, none of the individual chronologies should be discarded by default. Merging the chronologies obtained using different techniques to obtain "master chronologies for all the SISAL records would lead to a bias towards certain age-modelling approaches. For example, age models created using linear interpolation (successful 403 times) or Bchron (successful 420 times) would usually have a larger weight in a master chronology than Oxcal (successful 106 times) or linear regression (successful 182 times). Unless there is other information that suggests one age model type is better than others for a specific entity, all the chronologies should be considered equally likely.

To quantify the degree to which multiple age modeling techniques may reduce temporal uncertainty, I recommend that this manuscript includes a plot of the average of the analytical error in a record versus the average SISAL 95% chronology error in a record (i.e. average of 5b). Is the fit to that scatter plot a 1:1 line? Or is there a systematic reduction in the error across many records in the database b/c of time periods like 3400-3550 BP in Figure 5? Or do problematic areas, like unresolved hiatuses, compensate the reduction in errors for when the age model is tightly constrained? This would be an enlightening plot.

We do not expect a systematic reduction of the error as a result of the age-depth model techniques used and the data shows that the SISAL uncertainties ultimately depend on the uncertainties of the original U-Th dates and their spread/consistency. In any case, however, this would depend on the robustness of the approach used treat the uncertainties in each age-depth model.

As suggested by the reviewer, we will add a plot of the average analytical error per record vs the average SISAL chronology uncertainty as attached. The accompanying text will be:

*"The published age-depth models of all speleothems are accessible in the original-chronology metadata table and our standardised age-depth models are available at the sisal-chronology table for 512 speleothems. Temporal uncertainties are now provided for 79% of the records in the SISAL database. This is a significantly larger number than in SISALv1b, where most age-depth models lacked temporal uncertainties. Most speleothem records show average U-Th age errors between 100-1,000 years (Figure 6), which are only slightly changed by using age-depth modelling software. Nevertheless, when comparing the mean uncertainties of the U-Th ages with those of their corresponding age-depth model, the slope between both parameters is smaller than one. This indicates that age-depth models tend to reduce uncertainties especially when dating errors are large while they increase uncertainties, when U-Th age errors are small."*

Please give more detail in the text of the principles used in your calculations for when the SISAL chronology decides that there is a hiatus in the record. While you reference Breitenbach, 2012, it would be good to review the guiding principles that SISAL is using in lines 83-86 in more detail to make the manuscript more self contained. Also, what happens if there is disagreement among the various techniques about a hiatus – how does SISAL decide on a 'yes' or 'no' to split the record? Does majority rule??

Our age-depth model calculations use the U-Th dates and the depths of the hiatuses as entered in the database and which were provided by the researchers that produced the raw data and/or their publications.

We do not decide in our workflow whether there is (or should be) a hiatus in a section and therefore, all AM approaches use the same input data. For clarity, the input variables for the sisal chronologies are: depths of dating samples and isotopes, U-Th corrected ages and their uncertainties (if used to create the original or published age model), depth information of hiatuses if applicable, information on whether the speleothem was actively growing when collected and the year of collection. We did not attempt to assess whether the information provided by the data contributors/publications was correct. However, together with experts from the SISAL age modelling group we have checked whether the dates and hiatuses were visually consistent with the rest of the data.

We believe there is a misunderstanding as lines 83-86 refer to age reversals that occurred during the construction of the sisal chronologies instead of hiatuses:

*"Major challenges arise through hiatuses (growth interruptions) and age reversals. In the classification of the reversals, we distinguish between tractable reversals (with overlapping confidence intervals) and non-tractable reversals (i.e., where the two-sigma-dating uncertainties do not overlap) following the definition of Breitenbach et al. (2012)."*

We will rephrase this paragraph to clarify this point:

*"Major challenges arise through hiatuses (growth interruptions) and age reversals. We developed a workflow to deal with records with known hiatuses that allowed the construction of age-depth models for 20% of the records with one or more hiatuses (Roesch and Rehfeld, 2019; details below for each age-depth modelling technique). Regarding the age reversals, we distinguish between tractable reversals (with overlapping confidence intervals) and non-tractable reversals (i.e., where the two-sigma-dating uncertainties do not overlap) following the definition of Breitenbach et al. (2012). Details such as the hiatus treatment and outlier age modification are recorded in a logfile created when running the age models. We followed the original author's choices regarding date usage. If an age was marked as "not used" or "usage unknown", we did not consider this in the construction of the new chronologies except in OxCal, where*

*dates with "usage unknown" were considered."*

Details on how each technique tackled the hiatuses are copied below (with their corresponding line numbers):

**Linear Interpolation:** "Hiatuses are modelled following the approach of Roesch and Rehfeld (2019), where rather than modelling each segment separately, synthetic ages with uncertainties spanning the entire hiatus duration are introduced for use in age-depth model construction. These synthetic ages are removed after age-depth model construction. " **(lines 100-104)**

**Linear Regression:** "If hiatuses are present, the segments in-between were split at the depth of the hiatus without an artificial age. " **(lines 110-111)**

**Bchron:** "Since Bchron cannot handle hiatuses, we implemented a new workflow that adds synthetic ages with uncertainties spanning the entire hiatus duration (Roesch and Rehfeld, 2019), as performed with linear interpolation, StalAge and our implementation of COPRA. " **(lines 116-118)**

**Bacon:** "The R package rBacon can handle both outliers and hiatuses and apart from giving the median age-depth model, (. . .)." **(lines 125-126)**

**Oxcal:** "OxCal can deal with hiatuses and outliers and accounts for the non-uniform nature of the deposition process (Poisson process using the P-Sequence command). " **(lines 134-136)**

**COPRA:** " (. . .) we implemented a new workflow in R that adds artificial dates at the location of the hiatuses and prevents the creation of age reversals (Roesch and Rehfeld, 2019) as done with linear interpolation, StalAge and Bchron. " **(lines 150-152)**

**StalAge:** "The StalAge v1.0 R function has been updated to R version 3.4 and the default outlier and reversal checks were enabled to run automatically. Hiatuses cannot be entered in StalAge v1.0, but the updated version incorporates a treatment of hiatuses based on the creation of temporary synthetic ages following Roesch and Rehfeld

(2019). " **(lines 160-163)**

ESSDD